# Development and psychometric properties of the Nursing Student Academic Resilience Inventory (NSARI): A mixed-method study

Tayyebeh Ali-Abadi[1], Abbas Ebadi [2,3], Hamid Sharif Nia[4], Mohsen Soleimani[5], Ali Asghar Ghods [5]*

**1** Student Research Committee, Semnan University of Medical Sciences, Semnan, Iran, **2** Behavioral Sciences Research Center, Life Style Institute, Baqiyatallah University of Medical Sciences, Tehran, Iran, **3** Nursing Faculty, Baqiyatallah University of Medical Sciences, Tehran, Iran, **4** Educational Development Center, Mazandaran University of Medical Sciences, Sari, Iran, **5** Nursing Care Research Center, Semnan University of Medical Sciences, Semnan, Iran

* aaghods@yahoo.com

**Data Availability Statement:** All relevant data are within the manuscript and its Supporting information files.

## Abstract

### Introduction

Resilience has been proposed as a suitable solution to better deal with nursing students in cases of challenges but the complex and multidimensional nature of resilience has made its measurement challenging. This study aimed to develop and validate a new inventory theory-driven labeled Nursing Student Academic Resilience Inventory.

### Methods

This study was performed with an exploratory sequential mixed-method design. In the qualitative phase of the study, individual interviews were conducted by including 15 participants to elicit the concept of resilience through purposive sampling. In the quantitative phase, psychometric analysis of the extracted items was performed using face, content, and construct validities (exploratory and confirmatory factor analyses) on a sample size of 405 nursing students. Besides, reliability has been tested using internal consistency and test-retest methods. According to the COSMIN standards, beside two important indicators of validity and reliability, responsiveness and interpretability were also considered.

### Results

A 6-factor structure (optimism, communication, self-esteem/evaluation, self-awareness, trustworthiness, and self-regulation) with 24 items were extracted in terms of the derived categories from the qualitative phase. In confirmatory factor analysis, the $\chi^2/df$ ratio was calculated as 2.11 for the NSARI six-factor structure. Suitable values were obtained for the goodness of fit indices (CFI = 0.904, AGFI = 0.885, IFI = 0.906, PCFI = 0.767, and RMSEA = 0.053). In the second-order factor analysis, AVE = 0.70 indicated the existence of both convergent and divergent validities. The Cronbach's alpha and omega coefficients were investigated as (0.66–0.78) and (0.66–0.80), respectively. The AIC was between 0.33 and

**Funding:** The authors received no specific funding for this work.

**Competing interests:** The authors have declared that no competing interests exist.

0.45 for all factors, which is an acceptable rate. Additionally, an intraclass correlation coefficient (ICC) was obtained as .903 for the whole instrument (CI .846- .946, P <0.0001).

## Conclusion

Multidimensional nature of resilience was supported through exploring its 6-factor structures in the nursing students' field. This tool also showed an acceptable validity and reliability for measuring resilience in the population of nursing students.

## 1. Introduction

Previous studies conducted in Iran showed that nursing students are affected by various stressors such as lack of effective communication between trainer and students, unclear training goals, lack of essential efficacy when attending the patient's bedside, fear of infectious diseases, assignments' overload, and dissatisfaction with their field of study [1–3]. Resilience has been suggested as a beneficial solution for better dealing with challenges in nursing students, which leads them to play their professional roles in the future [4].

Resilience has evolved in the 1970s to understand why psychopathology is not always the consequence of risky environments among children [5]. Resilience is described as a characteristic, process or outcome that depends on the theory accepted by the researcher [6]. Theories can be effective on identifying and understanding the factors influencing an event or behavior as well as on determining how they work together. Therefore, a conceptual model can lead to obtain a better understanding of the existing conditions [7]. Resilience's theories can be outlined to the following three different models of resilience: "compensatory", "protective", and "inoculation/challenge" [5]. In this study, Stephen's resilience model as a protective model was used [4]. Whilst this model requires further empirical testing methods to validate its utility, it would provide a conceptual framework for designing a tool in nursing students, because this model mostly emphasizes on the importance of defining the concept of resilience as a process performed based on antecedent/perceived adversities, attributes/protective factors, and consequences/cumulative successes. Moreover, the proposed model combines perceived adversities with the use of individual protective factors to cope and/or be adapted with the difficulties effectively. The cumulative successes of these events would lead to the increased resilience demonstrated by the enhanced coping/adaptive abilities and well-being status [4].

The result of a review study yielded by Chmitorz et al. showed that various resilience scales have been developed [8], because resilience is a multidimensional construct that differs in different contexts, times, ages, and life conditions [9]. Several studies have found different factor structures of resilience in different populations and cultures [10, 11]. In addition, only one exploratory study was found performed in nursing student setting [12]. Therefore, it is not yet clear which factors measure the resilience in nursing students [13] that seems to be related to different natures of potential risk factors and protective processes which have led to some contradictions while the evaluation of designed interventions to promote resilience demands reliable and valid measures. In this way, researchers and clinicians do not have strong evidences regarding the choice of resilience measurement tools, which may make inappropriate choices in the field of study [14].

## 1.1 Research purpose and specific questions

To address the explained concerns, and regarding the lack of specific resilience tools in nursing students, the present study, in the first phase, aimed to explore the dimensions of resilience in nursing students based on the Stephen's resilience theory. Moreover, the overall aim of the study was to design a new valid and reliable inventory for measuring resilience in nursing students. Therefore, to achieve this aim, there were two research questions as follows:

1. What factors are related to the resilience of nursing students?

2. Which one of these factors has a statistically significant effect on the resilience of nursing students?

## 2. Methods

### 2.1 Study design and procedures

This exploratory sequential mixed-method study [15] was conducted in Iran from May 2019 to August 2020. It is assumed that all methods in this regard have bias and weaknesses, but based on triangulating data sources in this kind of design as well as the collection of both quantitative and qualitative data, the weaknesses of each form of data is neutralized [15]. The data in the qualitative phase were collected through performing semi-structured interviews based on the resilience Stephen's model in nursing students. Item generation and developing inventory were also done based on the categories extracted from the qualitative phase. Finally, psychometric properties were evaluated in the quantitative phase.

### 2.2 Participants

In the qualitative phase, thirteen under graduated bachelor's nursing students aged between 18 and 25 years old (5 female and 8 male), one female nurse, and one female trainer who had good communication skills, including being fluent in Persian language, no hearing or speech problem, and intended to participate in this study were recruited. Two students were dropout of keeping the education and one student was conditional student.

As well, the quantitative phase included several stages. Face validity was tested by 13 nursing students (7 female, 6 male) in different semesters. Moreover, 11 experts in instrument development, nursing, and psychology were invited by email to validate this inventory in terms of content validity. 200 nursing students for EFA as well as 205 nursing students for CFA analysis were recruited from 6 different provinces in Iran (age = 21.70 ± 2.50; female = 258, male = 147).

### 2.3 Data collection

In the qualitative phase, the semi-structured individual interviews were conducted in a quiet room in the university or the educational class of hospital regarding the student's choice. Students were selected through purposive sampling. The interviews were started by asking an ice-breaker question through which the individual was allowed to talk openly about the topic. Exploratory questions were asked following asking the main questions derived from the Stephen resilience model. The exclusion criteria were any circumstance that may interfere with study participation such as speech and hearing impairments and having work experience in medical settings. In the qualitative studies, sample size cannot be pre-established, so the interviews were perpetuated until data saturation [16].

Data from the qualitative phase were used to develop Nursing Student Academic Resilience Inventory (NSARI) for the quantitative phase. At face validity stage the new designed inventory were completed by nursing students. Concerning content validity all experts were expertise in tool's validation. Item analysis has been done by 36 nursing students who were selected using the stratified sampling. For construct validity, the final designed questionnaire was sent by some messengers to 405 nursing students who were selected by simple sampling method.

## 2.4 Statistical analysis

In qualitative phase, the analysis method used was deductive content analysis approach, which was recommended firstly by Elo et al. [17]. The interviews were audio-recorded, transcribed, and finally analyzed using MAXQDA software version 10. Data analysis was performed at three stages of preparation, organization, and reporting. During the preparation phase, semantic units were identified based on the purpose of the study. To get immersed in the data, the written interviews were read several times. To make sense, the researchers frequently asked wh-questions during performing the analysis. In the organizing phase, the text was read line by line and paragraph by paragraph in terms of the purpose of the study, and then important sentences and codes were identified. After this open coding, the lists of categories were classified under higher-order headings based on the comparison between these data and other observations that were not associated with the exact category. To assure data credibility, the samples were selected with maximum variations (from private and public universities, at different semesters, different ages, sexes, and scores). Additionally, to cover the variation of the phenomenon, a small data batch were firstly analyzed carefully and then it was determined that what additional data are needed in this regard. As well, peer checking was also done by three co-authors experienced in qualitative studies [17]. An example of the data analysis is shown in Table 1.

In the quantitative phase, the psychometric properties of the NSARI were assessed using face, content, and construct validities (exploratory and confirmatory factor analysis).

**2.4.1 Face and content validity.** Face validity only refers to the appearance of the instrument to the responders; in other words, what the test constructor intended to measure [18]. Confusing items were revised, duplicate, redundant items were merged, and essential items were added based on the results of the face and content validities. In terms of qualitative face validity, 13 nursing students were requested through convenience sampling to read the items loudly, explain the meaning of each item, and then identify problematic or ambiguous words for reconsideration. Following revising the items of the designed inventory according to the comments of the participants, at the qualitative face validity stage, quantitative face validity

**Table 1. An example of analysis process with category and theme.**

| Meaning unit | Condensed meaning unit | Code | Category | Theme |
|---|---|---|---|---|
| The higher experience, the more learning you will achieve. You can revise your mistakes and enhance your strengths. | Learning through doing more | Experience to find challenges | The enhanced coping /adaptive abilities | |
| Whatever you do for the second time, you will do it more safely if you have a successful previous experience. | Convenience after a successful experience | reducing stress through Successful experience | | Cumulative success |

was assessed using the impact factor method based on the following equation:

$$Impact\ Score = Frequency(\%) * Importance$$

The items equal to 1.5 and above were retained and the other items were modified [16, 19]. Due to the COSMIN definition, content validity is "the degree to which the content of an instrument is an adequate reflection of the construct to be measured" [16]. In this way, content validity ratio (CVR) was done by 11 experts to determine the essential items according to the cut-off point proposed in a study by Lawshe [19]. The Content Validity Index for each item (ICVI), and then the modified kappa coefficient were calculated based on the following equation [20]. Finally, S-CVI was estimated.

$$K = \frac{I.CVI - PC}{1 - PC} \rightarrow PC = \left\lfloor \frac{N!}{A!(N-A)!} \right\rfloor \times 0.5^N$$

**2.4.2 Item analysis.** Using a pilot study, the internal consistency was assessed before conducting the construct validity assessment, in order to recognize potential problems in the NSARI through estimating Cronbach's alpha and inter-item correlation. The items whose corrected item-total correlation score was less than 0.3, were then removed from the analysis [21].

**2.4.3 Construct validity.** *2.4.3.1. Exploratory factor analysis*. Exploratory factor analysis was conducted to test the factor structure of the items of the NSARI. Thereafter, the Kaiser-Meyer-Olkin (KMO) statistic of sampling adequacy and the Bartlett test of sphericity were calculated to check the suitability of the data for factor analysis. In this regard, KMO 0.8 was considered as acceptable. Hence, the latent factors were extracted by estimating the maximum likelihood; an oblique factor rotation technique, promax. The allocation of the item to a factor was determined based on the formula CV = 5.152 $\sqrt{}$ (n -2), which was obtained as approximately 0.3 [22]. According to the three indicator rules, at least 3 items must be retained for each factor in the EFA [23]. Finally, the items with communalities less than 0.2 were removed from the EFA [24].

**2.4.4 Confirmatory factor analysis.** Confirmatory factor analysis was done on different samples including 205 nursing students selected by convenience sampling, in order to examine fit indices of the extracted factors using Amos 24 software. Table 2 shows the accepted fit indices (CMIN/ DF, RMSEA, PCFI, IFI, CFI, and PNFI) [25]. In the second-order factor analysis, it was supposed that the extracted latent factors in the first-order factor analysis are a reflection of a more general concept in the upper levels [26]; therefore, the second-order factor analysis was performed following performing the first-order factor analysis.

**2.4.5 Convergent and divergent validity.** The criterion of Fornell-Larcker was used to estimate the convergent and divergent validities based on the average variance extracted (AVE) and the maximum shared variance (MSV). Of note, the convergent validity is generated when AVE> 0.5 and divergent validity is confirmed when MSV <AVE [27]. Additionally, Heterotrait-monotrait ratio of correlations (HTMT) criterion was clculated. It was established that all values in the HTMT matrix table should be less than .85 [28].

**2.4.6 Reliability, responsiveness, and interpretability.** Internal consistency of the NSARI was estimated using coefficients of Cronbach's alpha, and Omega McDonald [29]. Those values above 0.6 were considered as acceptable [30, 31]. AIC above 0.2–0.4 was acceptable [32]. Construct reliability, which replaced Cronbach's alpha in the structural equation model, was considered as acceptable in all values above 0.7 [33]. Test-retest reliability was conducted to investigate the questionnaire's stability, in a way that 36 nursing students were requested to fill the questionnaire twice with a two-week interval. In this study, responsiveness

**Table 2. Exploratory factor analysis of the NSARI.**

| Factor | Items | Factor loading | $h^{2\,a}$ | Eigenvalue | Variance (%) |
|---|---|---|---|---|---|
| **Optimism** | **Q12**. There are many opportunities in nursing field for me | .789 | 0.58 | 2.82 | 11.75 |
| | **Q20**. I am looking forward to be a great nurse in near future. | .739 | 0.58 | | |
| | **Q21**. I am looking forward to an economic prosperity in the field of nursing. | .719 | 0.37 | | |
| | **Q17**. I have a positive attitude toward nursing. | .712 | 0.56 | | |
| | **Q29**. I am motivated to participate in internships. | .360 | 0.36 | | |
| **Communication** | **Q24**. Patient companion reduces my stress. | .739 | 0.47 | 1.97 | 8.2 |
| | **Q26**. Understanding my terms by my instructor, reduces my stress | .641 | 0.49 | | |
| | **Q25**. Cooperation by nurses reduces my stress. | .584 | 0.38 | | |
| | **Q23**. Having a good communication with attending physicians reduces my stress. | .420 | 0.40 | | |
| **Self-esteem/ evaluation** | **Q5**. My instructors trust my judgment in taking care of the patients. | .706 | 0.56 | 1.53 | 6.4 |
| | **Q4**. I get support from my instructors. | .643 | 0.38 | | |
| | **Q10**. I have sufficient confidence in taking care of patients. | .344 | 0.44 | | |
| | **Q15**. I have adequate knowledge in (science of) nursing. | .316 | 0.29 | | |
| **Self-awareness** | **Q9**. I manage the difficulties of my academic years. | .825 | 0.62 | 1.58 | 6.6 |
| | **Q11**. I'm not disappointed by the failures during my education. | .521 | 0.45 | | |
| | **Q30**. I have adequate motivation to participate in theory sessions. | .365 | 0.36 | | |
| **Trustworthiness** | **Q2**. I earn my patient's trust by making a suitable communication. | .570 | 0.48 | 1.41 | 5.9 |
| | **Q1**. By strengthening my nursing knowledge, I will do my best to take care of the patient. | .470 | 0.50 | | |
| | **Q6**. I get support from my family. | .439 | 0.34 | | |
| | **Q7**. My friends and colleagues support me. | .420 | .31 | | |
| **Self-regulation** | **Q14**. I examine all options to reach my goals. | .783 | 0.43 | 1.59 | 6.62 |
| | **Q22**. My attempts are to strive and reach my goals. | .461 | 0.41 | | |
| | **Q32**. I learn better by executing bedside procedures. | .393 | .36 | | |
| | **Q13**. I try to endure the academic hardship. | .336 | .37 | | |

a. Communalities.

was also determined by the Standard Error of Measurement (SEM) and the Minimal Detectable Change (MDC) score [34]. The MDC less than 30% was acceptable, and below 10% was considered as excellent [35]. To determine the interpretability of the NSARI, the distribution of total scores in the whole samples, floor, and ceiling effects was calculated. It was considered that the floor and ceiling effects are present if more than 20% of respondents attained the lowest or the highest score, respectively [36]. All the statistical analyses were done by SPSS-AMOS25 and SPSS R-Menu v2.0.

**2.4.7 Normal distribution, outliers, and missing data.** The univariate and multivariate normal distributions of data were investigated by skewness (±3) and kurtosis (±7), respectively [25]. The presence of a multivariate outlier was assessed by Mahalanobis d-squared ($p < .001$), and multivariate normality was assessed by Mardia coefficient of multivariate kurtosis ($< 20$) [37]. There was no missing data in this study due to completing the surveys by Porsline. In this online survey software, responders must respond to all questions because following and responding one question depends on answering the previous one.

## 2.5 Trustworthiness in the mixed-method study

To develop self-reflexivity, after transcribing each one of the interviews, the researchers analyzed the obtained data to uncover their preunderstandings. As well, the researchers consulted

with team members on the extracted codes and themes and then described the data analysis process in detail and finally provided clear citations in this regard.

To diminish the threats to the internal and external validities of the mix-method study, the participants with different experiences were selected. As well, none of the samples in the qualitative phase participated in the quantitative phase. Furthermore, designing the item pool was performed based on the main categories and sub-categories extracted in the qualitative phase. Consequently, all the stages of the study were carefully reviewed and verified by the other researchers [19, 38].

### 2.6 Ethical consideration

This study was approved by the ethics committee of Semnan University of Medical Sciences (Approval Number IR.SEMUMS.REC.1398013) to develop a new valid and reliable tool to measure resilience in nursing students. The researchers included in this study were not working at the same institutions. Therefore, there was no conflict of interest. The researcher referred to the education deputy and received the demographic list of the included students. In several sessions, the students (in different educational semesters) were invited to participate in the study. Thereafter, the aim of this study, data confidentiality, and voluntarily participation were explained to them. In qualitative phase, the obtained informed consent was verbal, because using this way; the participants felt more intimate relationship for participating in their interview sessions. The IRB at this site was inclined to this procedure. Thereafter, in the quantitative phase, surveys were completed through Porsline software. At the beginning of the inventory, written informed consent was attached and the subjects declared their consent by clicking on "I agree to participate".

## 3. Results

### 3.1 Qualitative phase

After condensing the codes extracted from the semantic units, 797 codes were finally obtained. These codes were allocated to three themes (Perceived stress, Protective factor, and Cumulative success), 9 categories (perceived stress: environment, relationship, Social-standing, Burnout, and moral distress; protective factor: personal and social; and cumulative success: Wellbeing and the enhanced coping /adaptive abilities), and 31 subcategories based on interviews in the field of nursing student and the Stephen's model. Moreover, an item pool consisting of 93 items was generated to explain the resilience of nursing students based on the directed content analysis. Following discussing with the research team, the number of the items was reduced to 83 items. Table 1 demonstrates how items were generated from the qualitative results.

### 3.2 Quantitative phase

The clarity of the items in the initial item pool was evaluated by 13 nursing students through qualitative face validity. Afterward, the results of calculating the impact score of each item showed that all the 83 extracted items are suitable for measuring resilience. According to the results of CVR, the items with a numerical value of less than 0.59 were removed, and 52 items remained at last. Thereafter, 11 experts in the fields of instrument development, nursing, and psychology were recruited to assess CVR and CVI. Finally, after removing the items with the kappa index less than 0.74, 45 items were sent to the nursing students for item analysis. Notably, SCVI inventory was obtained as 0.92.

**3.2.1 Item analysis.** Cronbach's alpha for 45 NSARI items was 0.88. Intra-class correlation coefficient of the NSARI total score specified a high level of temporal stability (*ICC* = 0.88, *95% CI*: 0.82–0.93). Finally, 13 items were eliminated from the analysis due to their corrected item-total correlation scores were less than 0.3.

**3.2.2 Construct validity.** Total of 405 participants with the mean age of 2.72± 21.67 years old were recruited for construct validity. The majority of the students were women (N = 258, 63.7%). The adequacy index of sampling was 0.892, and the Bartlett's test was statistically significant (df = 325, χ2 = 3123.231, P< 0.001).

*3.2.2.1. Exploratory factor analysis.* During performing exploratory factor analysis, 8 items were removed due to their low factor loading. The remaining 24 items were then loaded on six factors, which finally explained 45.47% of the total variance (Table 2). Correspondingly, the first factor was optimism, which can be defined as people's positive expectations of what is happening in their lives [39]. In addition, having a positive optimistic attitude can help in more effectively dealing with stress [40]. The second factor was labeled as communication defined as the exchange of information, ideas, and feelings among people who use speech or other means [41]. Self-esteem/ evaluation as the third factor is known as a prime predictor of stress management indicateingthe degree of belief in the ability, importance, success, and individual's competency [42]. The fourth factor was identified as self-awareness. Accordingly, this psychological component implies awareness of feelings, motivations, self-concept, and personality [43]. Trustworthiness as a worth demanding nurses to foster certain character attitudes and to strength effective task performance [44], was characterized as the fifth factor. Finally, self-regulation as the sixth factor was defined as the student's attempt to manage learning processes focused on fulfilling goals [45].

*3.2.2.2. Confirmatory factor analysis.* Table 3 represents the Chi-square test and other fitness indices for the first-order CFA compared to the second-order model (Table 3).

According to the final factor structure of the NSARI, some correlations between the measurement errors of items 4 and 5, 11 and 30, 6 and7, and 12 and 29 were detected (Fig 1). The second-order factor analysis was conducted to investigate whether all the factors fitted the general concept of "Resilience". Fig 2 shows the structural model as well as the CFA of the NSARI.

**3.2.3 Convergent and divergent validities and internal consistency.** The convergent and divergent validities of the NSARI were not confirmed in the first-order factor analysis due to the presence of a latent factor, which indicated that all the extracted factors were a reflection of the concept called "Resilience". The results of the second-order factor analysis (AVE = 0.70)

**Table 3. Fitness indices in the first and second-order factor analysis.**

| Indexes | Cut-off values | First-order | Second-order |
|---|---|---|---|
| **CMIN/DF** | < 3 | 2.11 | 2.23 |
| *P*- value | ≥ 0.05 | <.0001 | <.0001 |
| **RMSEA** | ≤ 0.08 | 0.053 | 0.055 |
| **PCFI** | ≥ 0.5 | 0.767 | 0.783 |
| **CFI** | ≥ 0.95 | 0.904 | 0.890 |
| **IFI** | ≥ 0.90 | 0.906 | 0.891 |
| **AGFI** | ≥ 0.90 | 0.885 | 0.875 |
| **PNFI** | ≥ 0.5 | 0.708 | 0.721 |

CMIN/DF: Minimum Discrepancy Function by degrees of freedom divided; RMSEA: Root mean square error of approximation; PCFI: Parsimonious Comparative Fit Index; CFI Comparative Fit Index; IFI: Incremental fit index; AGFI: Adjusted Good of Fit Index; PNFI: Parsimonious Normed Fit Index.

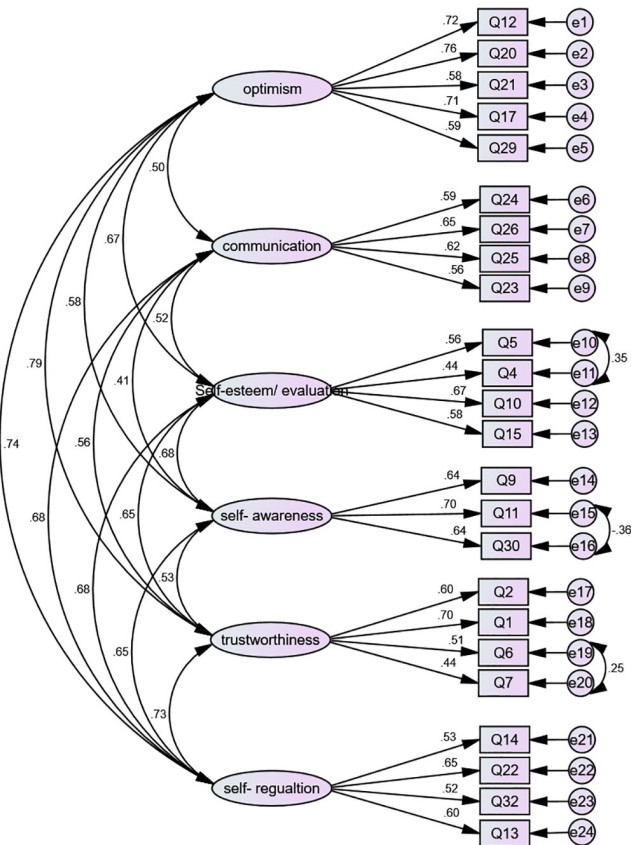

**Fig 1. The first-order factor analysis of the NSARI.**

indicated the existence of convergent and divergent validities. As well, the value in the HTMT matrix was less than .85, indicating that discriminat validity was established in this study (Table 4).

The internal construct reliability was confirmed using cronbach's alpha coefficients and McDonald omega. The AIC was obtained between 0.33 and 0.45 for all the factors, which was acceptable (Table 5). Moreover, the ICC was calculated as 0.903 (CI .846- .946, P <0.0001) through the test-retest method for the whole inventory.

**3.2.4 Responsiveness and interpretability.** The minimum detectable change percentage was calculated as below ten percent (8.61%) and the standard error of measurement was also determined (3.12). The effect of ceiling and floor on the total inventory scores was estimated to be less than 20%. Additionally, the mean and standard deviation of resilience score was different in students from different gender, with high and low GPA, semester, and different ages. The results of t-test and ANOVA showed that all these variables, except gender and age variables, had a significant difference between resilience scores (Table 6). Tukey post hoc test's result about semester revealed that there was a significant difference between the second and sixth semesters (P = 0.02).

**3.2.5 Scoring items.** Each item was rated on a five-point Likert scale ("completely agree", "agree", "no idea", "disagree", and "completely disagree"). The scores of the inventory were from zero to one hundred, and then the following scores were transformed to standard scores

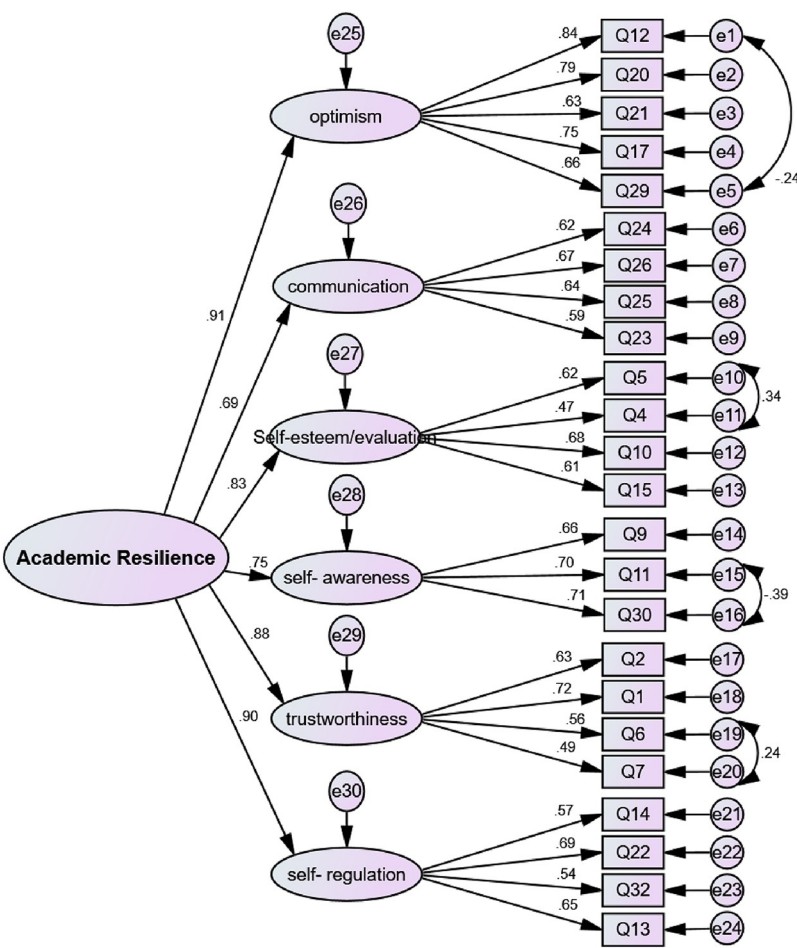

**Fig 2. The second-order factor analysis of the NSARI.**

through the linear scoring using the formula stated below:

$$Transformed\ scale = (\frac{The\ obtained\ row\ score - The\ lowest\ possible\ raw\ score}{The\ highest\ possible\ raw\ score - The\ lowest\ possible\ raw\ score}) * 100$$

Obviously, after converting the scores to the standard values, the higher average score of resilience close to one hundred indicated the higher score of the resilience in nursing students.

**Table 4. Discriminant validity assessment using the HTMT matrix.**

| | | Factor 1 | Factor 2 | Factor 3 | Factor 4 | Factor 5 | Factor 6 |
|---|---|---|---|---|---|---|---|
| **Heterotrait-monotrait ratio of correlations (HTMT)** | **Factor 1** | | | | | | |
| | **Factor 2** | 0.525 | | | | | |
| | **Factor 3** | 0.641 | 0.532 | | | | |
| | **Factor 4** | 0.639 | 0.465 | 0.685 | | | |
| | **Factor 5** | 0.754 | 0.517 | 0.678 | 0.614 | | |
| | **Factor 6** | 0.722 | 0.694 | 0.629 | 0.696 | 0.706 | |

**Table 5. Internal consistency and construct reliability of NSARI.**

| Factor | α | AIC | Ω | CR |
|---|---|---|---|---|
| Optimism | 0.787 | 0.450 | 0.806 | 0.806 |
| Communication | 0.678 | 0.367 | 0.701 | 0.700 |
| Self-esteem/evaluation | 0.685 | 0.357 | 0.691 | 0.653 |
| Self-awareness | 0.631 | 0.371 | 0.664 | 0.700 |
| Trustworthiness | 0.647 | 0.342 | 0.675 | 0.653 |
| Self-regulation | 0.662 | 0.330 | 0666 | 0.663 |

CR: Construct Reliability, AIC: Average inter item correlation.

**Table 6. Distribution of resilience scores in nursing students.**

| Variable | | Mean± SD | Result |
|---|---|---|---|
| Semester | 1 | 31±116.88 | F = 3.29; df = 7; p = 0.002 |
| | 2 | 16.27±128.31 | |
| | 3 | 17.29±122.48 | |
| | 4 | 14.44±122.52 | |
| | 5 | 11.53±130.28 | |
| | 6 | 16.16±120.25 | |
| | 7 | 14.73±133.55 | |
| | 8 | 16.87±124.79 | |
| GPA | <17 | 16.92±123.32 | t = -3.10; df = 382; p = 0.002 |
| | >17 | 14.56±130 | |

# 4. Discussion

The aim of present study was to develop a new inventory for measuring the resilience in nursing students. Although numerous tools have been designed in different contexts, there is only one study designed in nursing students. The Yang et al.'s study extracted 24 items through literature review and student interviews in two personal and contextual dimensions among Korean nursing. These dimensions were self-confidence, positivity, coping ability, emotional regulation, structured style, relationship, and social support [12]. As shown in this study, resilience is a multidimensional concept, but some other dimensions such as relationship and being able to cope are the core of the resilience. These results of this study support the result of the present study. Moreover, Thomas et al. in their integrative review declared some factors affecting resilience in nursing students that were grouped into the following three themes including support, time, and empowerment [46].

In this study, the NSARI including 24 items was designed. The psychometric of the NSARI showed the validity and reliability of the inventory. Furthermore, its internal consistency detected that all the items could measure resilience. Regarding reproducibility, the result of ICC identified that this inventory can probably catch up with reliable results at different times and places. The Cronbach's alpha coefficient was obtained to be the range of 0.63 to 0.78. Nunnally et al. [31], and Moss [30] in their studies reported 0.6 as an acceptable value [31].

Based on exploratory factor analysis, six-factors, including optimism, communication, self-esteem/evaluation, self-awareness, trustworthiness, and self-regulation were extracted, which

explained 45.47% of the total variance. Similarly, another review literature showed the existence of instability in the factor structure of the resilience scales in different populations [47]. The first-factor extracted in the present study was optimism. Numerous studies have shown the relationship between optimism and students' ability to adapt, besides their educational performance. Soares et al. declared that a low level of optimism is a predictor of some problems such as maladaptation with the academic environment, the incidence of depressive symptoms, isolating feeling, stress symptoms during the first year of the course, and the reduced academic performance in subsequent years [39]. In another study, Camp showed that the academic optimism is positively correlated with surviving till finals and becoming well-qualified in the field, motivation, and academic achievement [43]. The second factor was communication. Accordingly, having the ability of an effective communication is an essential element for nursing students to feel well-being, which subsequently increases the level of their self-confidence, motivation, and self-esteem [48]. Also, Sheu et al. reported that the inability to communicate with nurses is one of the most important stressors among senior students [49]. Additionally, academic competency is evaluated based on academic achievement, classmate relationship, and social behaviors [50]. Furthermore, Marañón in their study stated that the student-instructor relationship is a determining factor in learning, which makes progress in the learning process, leads to shape students' identities as forthcoming nursing professionals [51], and increases their motivation for learning [52]. The third extracted factor was self-esteem/evaluation. Having a high-level of self-esteem in managing the needs of nursing students during the practice course, as well as establishing a strong and therapeutic relationship with the patient is crucial [42]. In the present study, the students attempted to achieve self-esteem through knowledge enhancement, and making trust and support. The fourth extracted factor was self-awareness. Accordingly, since self-awareness is a key for preventing unhealthy reaction to stress, so nursing has accepted Maslow's theory of motivation and Rogers's view of self-awareness as fundamental issues of professional nursing [53]. This was also mentioned as an antecedent to high-quality nursing care [54], so that awareness of identity as a nurse allows us to be aware of the positive and negative effects of our roles. By being aware of the effects of characteristics such as age, gender, and race on another person, the nurse can better perform her/his job. As well, the "nursing process" is often described as a problem-solving method and the nurse's self-awareness will help in answering the question what is the problem?. On the other side, by understanding the effects of characteristics such as age, gender, and race on another person, a nurse can better gain the patient's trust and cooperation to stimulate patients' active participation [55]. Next, the fifth extracted factor was trustworthiness. The development of resilience in nursing students hinges on a trust-based educational culture, which paves the way for caring. Nursing students can better focus on the needs of the patient in this culture [56]. In a systematic review of qualitative studies, Rørtveit et al. stated that nurses who facilitate the trust relationship, comprehensively engage, listen, and act as patient supporters [57]. In this study, total students tried to meet the patient's needs by establishing a trust-based relationship through strengthening nursing knowledge. To achieve this goal, they benefited from the support of their family and friends. The last extracted factor was self-regulation. One of the protective factors in resilience is the development of self-regulation (both emotional and behavioral regulations) [50]. Indeed, those students who personally adjust their learning process, will learn more effectively [58]. Self-regulation is a deed that includes determining learning goals, adjusting personal effort, participating in time management, supervision, and evaluating existing performance [59]. In addition, one must be directed to a specific goal or organization to be motivated [60]. In this way, achievement in theoretical courses and becoming an expert in the clinical settings were considered as the goals of nursing students [45]. Therefore, students mostly try to maintain and promote resilience by determining and achieving compeling and

task-based goals. In addition, according to the practical identity of nursing, the more the students do clinical procedures, the more expert they become.

Regarding the modification indices, 4 pairs of measurement errors among the measured items of the first, third, fourth, and fifth factors were allowed to co-vary freely, which consequently improved the final model fit noticeably.

According to the final model of NSARI, there was a correlation among the measurement errors of some items. The correlated measurement error occurs in the situation where variables were not recognized apparently or not measured directly. In addition, measurement errors may be resulted from the use of self-report data [61].

Construct validity indices showed the convergent and divergent validities in the final model. Convergent validity exists when the factors of the instrument are close together and explain a large variance. In other words, divergent validity exist when the extracted factors are separated from each other [37].

### 4.1 Limitations

Despite the validity and reliability of the inventory and the final approval of the factor structure, the present study has some limitations, including the method of data gathering through the self-report method and the sample disproportion regarding gender; however, this is common because nursing is a female-dominated profession [62]. Also, the design of the study was cross-sectional, so therefore to study the effects of time, and individuals' characteristics, follow-up studies will be valuable to investigate the changes in the resilience factor structure. In addition, the percentage of extractive variance was calculated almost less than 50%, which is accepted by many psychometric studies. Moreover, a number of fit indices were found to be under the recommended cut-off point. To the best of our knowledge, the NSARI is the first inventory to measure resilience in a theoretically driven way that has the potential to track and predict resilience in nursing students, because of evidences of validity and reliability in this population. To ensure that the instrument is psychometrically robust, its psychometric properties should be tested using different statistical methods such as the Rasch model and structural equation modeling, in order to estimate age and gender effects. Further studies are recommended to investigate the conceptual structure of this inventory to gather more evidences regarding the psychometric properties of the inventory study.

## Supporting information

**S1 Dataset.**
(SAV)

## Acknowledgments

We appreciate all nursing students who participated in this study, and also all the tutor and nurses who helped us to carry out this study.

## Author Contributions

**Conceptualization:** Tayyebeh Ali-Abadi.

**Data curation:** Tayyebeh Ali-Abadi.

**Formal analysis:** Tayyebeh Ali-Abadi, Ali Asghar Ghods.

**Methodology:** Tayyebeh Ali-Abadi, Abbas Ebadi, Hamid Sharif Nia, Mohsen Soleimani, Ali Asghar Ghods.

**Software:** Abbas Ebadi, Hamid Sharif Nia, Ali Asghar Ghods.

**Supervision:** Abbas Ebadi, Ali Asghar Ghods.

**Validation:** Abbas Ebadi, Hamid Sharif Nia, Ali Asghar Ghods.

**Writing – original draft:** Tayyebeh Ali-Abadi.

**Writing – review & editing:** Abbas Ebadi, Hamid Sharif Nia, Mohsen Soleimani, Ali Asghar Ghods.

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
