## [Decision Letter · Decision Letter 0]

14 Jan 2021

PONE-D-20-37513

Development and psychometric properties of the Nursing Student Academic resilience Inventory (NSARI): A Mixed-Method study

PLOS ONE

Dear Dr. ghods,

Thank you for submitting your manuscript to PLOS ONE. After careful consideration, we feel that it has merit but does not fully meet PLOS ONE’s publication criteria as it currently stands. Therefore, we invite you to submit a revised version of the manuscript that addresses the points raised during the review process.

We encourage the Authors to accomplish the Reviewers' requests and suggestions as much as possible and/or to give reasonable justifications for modifications they consider not suitable to make.

We look forward to receiving your revised manuscript.

Kind regards,

Paola Gremigni, Ph.D.

Academic Editor

PLOS ONE

Journal Requirements:

2. Please include additional information regarding the questionnaires and qualitative guides used in the study and ensure that you have provided sufficient details that others could replicate the analyses. For instance, if you developed a questionnaire as part of this study and it is not under a copyright more restrictive than CC-BY, please include a copy, in both the original language and English, as Supporting Information, or include a citation if it has been published previously.

3.We suggest you thoroughly copyedit your manuscript for language usage, spelling, and grammar. If you do not know anyone who can help you do this, you may wish to consider employing a professional scientific editing service.  

4.Thank you for stating the following financial disclosure:

6. Please include a separate caption for each figure in your manuscript.

Additional Editor Comments:

Dear Authors,

Four Reviewers have completed the review of the manuscript. Most of them do not consider the manuscript acceptable in its actual form; however, I encourage the Authors to accomplish as much as possible the Reviewers' requests and suggestions and resubmit a new version of their work.

Reviewers' comments:

Reviewer's Responses to Questions

**Comments to the Author**

1. Is the manuscript technically sound, and do the data support the conclusions?

Reviewer #1: Yes

Reviewer #2: Partly

Reviewer #3: Partly

Reviewer #4: No

2. Has the statistical analysis been performed appropriately and rigorously? 

Reviewer #1: I Don't Know

Reviewer #2: I Don't Know

Reviewer #3: No

Reviewer #4: No

3. Have the authors made all data underlying the findings in their manuscript fully available?

Reviewer #1: Yes

Reviewer #2: Yes

Reviewer #3: Yes

Reviewer #4: Yes

4. Is the manuscript presented in an intelligible fashion and written in standard English?

Reviewer #1: Yes

Reviewer #2: No

Reviewer #3: No

Reviewer #4: Yes

5. Review Comments to the Author

Reviewer #1: Dear Author,

1. It is better to write Sample size same in the abstract (15) and the main study (13).

2. If you write the Abbreviations (minimum) in the study readers can read easily your article.

3. You write the conclusion section in the abstract. It is better to write in the main study although it is optional (journal requirement)

4.Exploratory factor of the NSARI seems very good.

5. in Table 2 better to write abbreviations below the table.

Reviewer #2: Overall:

The topic is an important one; the instrument at face value has promise though there is a lack of discussion about what gaps exist in current instruments that this instrument addresses. Developing the instrument based on a model strengthens validity. It is clear that the authors have done a significant amount of work. The manuscript content covers a large amount of information and the reviewer wonders if this work would better be served by breaking it up into two manuscripts. The reviewer is concerned that, due to word/space limits, study details are not able to be described adequately. There are multiple areas where additional information is needed to fully convey study details and supportive information.

Concerns:

Grammar, word choice & sentence structure issues are barriers to fully comprehending the manuscript content. Most of the manuscript is also written in present/future tense; if the studies are completed, it should all be written in past tense.

There is not enough detail about current instruments and what makes this instrument necessary.

P. 7 lines 150-151: the authors state “There is no missing data in the study due to completing the surveys by an online Survey software (Porsline).” The reviewer does not understand how having surveys completed online assures no missing data.

Description of instrument development is challenging to follow and to comprehend.

A description of the factors comes in the discussion section, but there is lacking a robust description of these in the results section.

The reviewer believes that content in this manuscript was ambitious and likely too much to include in a single manuscript. Ultimately, this detracted significantly from fully comprehending and appreciating the processes of instrument development and the value of this newly developed instrument.

Reviewer #3: Thank you for the hard work on this manuscript. The authors did well by performing different analyses to establish the validity and reliability of the tool. Please attend to the following

1.kindly check the grammar and ensure correct sentence structures

2.You indicated using Stephen's Resilience Model. This is a good model but I don’t see from your writing how this model informed your research.

3.The title and introduction look as if you are developing an entirely new tool. However, the factor structures seem to suggest that you were validating the NSARI tool in the Iranian context. Can you please clarify this?

4.The qualitative results did not indicate which items were generated and how they were elucidated from the qualitative study. It will be good to get a clear path on how items are generated from the interview data before psychometric analysis. At the moment, your qualitative results did not capture that. It is advisable to, at least, state the themes, categories, and sub-categories backed by participant quotes. This will provide evidence on how you arrived at the items that are used in the quantitative phase

5.In line 132, you indicated the inventory was revised by 13 nursing students in the qualitative face validity stage. How were these revisions done because your qualitative methodology only talk about how the items would be developed, and not about revision

6.Under the results in the quantitative phase, Line 209-2016 seems to be talking about method rather than results. Can you move this to the methods section?

7.In line 209-210, you indicated that “In the psychometric evaluation phase, confusing items are revised and duplicate, redundant items are merged and essential items are added based on the results of the face and content validity” How were these revisions and merging done. Where from the ‘essential items”? Face validity, as we know is usually done by experts. Who were the ‘experts’ in your study? Please kindly explain how this was done as it forms an essential aspect of instrument development

8.I am not too conversant with your measurement of convergent and divergent validity. However, I do know that the common, and probably the best, way of determining divergent validity for instance is to measure the focal construct (in this case resilience) and another closely related concept using the same instrument. Divergent validity would be established when the two measures are weakly correlated. This is to ensure that the tool is not measuring other constructs. I know this may not have been your intent but it is worth knowing.

Reviewer #4: Dear Authors,

I had the opportunity to review the manuscript "Development and psychometric properties of the Nursing Student Academic resilience Inventory (NSARI): A Mixed-Method study" for PLOS ONE. In my review of the paper, I will provide suggestions that may be useful for these authors to take into account in future studies and/or revisions of this manuscript.

One of the main comments is related to lack of flow in reading in the methods session. Perhaps creating a specific section for the detailed description of the participants, another session for the details of the procedure, another session for the measurements and finally statistical analysis, could improve the reading considerably.

For construct validity, the authors used a methodological approach that is commonly used in the field; however, the approach assumes a linear relationship between the items and the factor. Also, this methodology assumes a gaussian distribution of the outcome. This approach can be a problem for the construction of domains. Accurate methods for detecting relation between items are relevant for designing and determine the factor distribution. I suggest using Item Response Theory (IRT), specifically Graded Response Model (GRM; Samejima, 1969, 1997), Generalized Partial Credit Model (GPCM; Muraki, 1992) or multidimensional graded response model (MGRM). The IRT models examine the overall response patterns across all the items, while factor analytic methods examine covariances between the individual items. As a consequence of evaluating item response patterns, the parameter estimates provide insight into how the items work. Finally, the IRT estimates constructs assuming nonlinear relationship between latent traits and item responses.

For IRT authors can use mirt package. It’s a powerful package and free tool for psychometric analyses: Chalmers, R. P. (2012). mirt: A multidimensional item response theory package for the R environment. Journal of Statistical Software, 48(6), 1-29.

Another suggested approach to construct validity is an exploratory structural equation modeling (see Asparouhov et al., 2009) approach.

Asparouhov, Tihomir, and Bengt Muthén. "Exploratory structural equation modeling." Structural equation modeling: a multidisciplinary journal 16.3 (2009): 397-438.

I’m agreed with authors, it’s necessary to estimate age and sex effect. However, I suggest reviewing the papers related to this issue of Rivera and Wim Van der Elst:

Rivera, D., Olabarrieta-Landa, L., Van der Elst, W., Gonzalez, I., Rodríguez-Agudelo, Y., Aguayo Arelis, A., ... & Arango-Lasprilla, J. C. (2019). Normative data for verbal fluency in healthy Latin American adults: Letter M, and fruits and occupations categories. Neuropsychology, 33(3), 287.

Van der Elst, W., Ouwehand, C., van Rijn, P., Lee, N., Van Boxtel, M., & Jolles, J. (2013). The shortened raven standard progressive matrices: item response theory–based psychometric analyses and normative data. Assessment, 20(1), 48-59.

Please add the exclusion criteria that were used to enroll participants in the study.

Please provide information about how the sample size was calculated in each group.

Provide more information about dace and content validity data analyses technique

6. PLOS authors have the option to publish the peer review history of their article (what does this mean?). If published, this will include your full peer review and any attached files.

Reviewer #1: No

Reviewer #2: **Yes: **Kristen Abbott-Anderson, PhD, RN, CNE

Reviewer #3: No

Reviewer #4: No

---

## [Author Response · Author response to Decision Letter 0]

2 Feb 2021

Dear Editor-in-Chief of PLOS ONE Journal

Dr. Joerg Heber

Response to reviewers

Thank you for your thorough review and consideration of our manuscript “Development and psychometric properties of the Nursing Student Academic resilience Inventory (NSARI): A Mixed-Method study" that was submitted to PLOS ONE Journal.

We believe that the comments and suggestions that were recommended by the reviewers have informed a much improved and more fully developed paper that will offer an important contribution to the field.

Given the word length requirements for clinical research briefs, we are somewhat stymied by the extent of details we can provide, However, and of critical importance, attention has been paid to developing a clearer rationale that addresses the significance of this work. More information has been provided to describe the measures used and data analysis conducted and this revised paper has more clarity about the comments and suggestions the reviewers has proposed. 

We have highlighted the changes that were undertaken in response to your comments in the revised manuscript. A response to each of your comments is below. 

Please feel free to contact me with any questions and concerns. I look forward to hearing from you regarding the manuscript.

Thanks for your kind attention to the manuscript.

Sincerely, Corresponding author

1. It is better to write Sample size same in the abstract (15) and the main study (13).Due to this study is a mixed method study and has qualitative and quantitative phase , also in quantitative phase has several stage so the number of participants is different in every stage but overall, the number of participants in the qualitative stage was 15 participant and in final stage of psychometric (EFA, CFA) was 405.

P: 2 Methods: individual interviews are conducted with 15 participants 

psychometric analysis …….. with a sample size of 405 nursing students.

2. If you write the Abbreviations (minimum) in the study readers can read easily your article.P:7,8,9,13

The abrevation are written, and highlited. 

3. You write the conclusion section in the abstract. It is better to write in the main study although it is optional (journal requirement).The abstract was written based on instruction for author.

4.Exploratory factor of the NSARI seems very good.Thank you for your kind comment.

5. in Table 2 better to write abbreviations below the table.P: 12.

It is written , and highlighted

Reviewer2

Grammar, word choice & sentence structure issues are barriers to fully comprehending the manuscript content. Most of the manuscript is also written in present/future tense; if the studies are completed, it should all be written in past tense.The present/future tense has been changed to past tense.

There is not enough detail about current instruments and what makes this instrument necessary.P: 4

Diverse approaches to measuring resilience affected by different nature of potential risk factors and protective processes have led to contradictions [8]. While establishing the resilience concept as a meaningful concept in research and clinical practice, it is inevitable to determine its distinctive factors and measure its factors in a valid and reliable method [9]. Different studies have found different factor structures of resilience tools in different populations and cultures [11, 12]. Even though measuring resilience has done by a number of researchers such as Charney[11], Wagnild[12], Kobasa[13], but researchers and clinicians do not have strong evidences about the choice of resilience measurement tools and may make inappropriate choices in the field of study [13]. Notwithstanding the prevalence of stress and the obligation to design an appropriate tool to assess students' resilience, it is not yet clear which factors measure the resilience in nursing students

P. 7 lines 150-151: the authors state “There is no missing data in the study due to completing the surveys by an online Survey software (Porsline).” The reviewer does not understand how having surveys completed online assures no missing data. In online Survey software responders have to respond all question because of following and responding one question was dependent on answering previous question, so much so,there was no missing data.

A description of the factors comes in the discussion section, but there is lacking a robust description of these in the results section. P:11 

Items 12, 20, 21, 17,29 were assumed to mention optimism, items 24, 26,25,and 23 were reflected communication, Self-esteem/ evaluation was attributed to items 5,4,10,15, also items 9,11,30 composed Self-awareness, items 2,1,6,7 formed trustworthiness, and items 14,22,32, and 13 were hypothesized to reflect the Self-regulation (Table 2).

The reviewer believes that content in this manuscript was ambitious and likely too much to include in a single manuscript. Ultimately, this detracted significantly from fully comprehending and appreciating the processes of instrument development and the value of this newly developed instrument.Due to avoiding salami publication, this article has been written as mixed method study.But we are somewhat stymied by the extent of details we can provide, However, and of critical importance, attention has been paid to developing a clearer rationale that addresses the significance of this work.

Reviewer3

1- kindly check the grammar and ensure correct sentence structures. It is done.

2- You indicated using Stephen's Resilience Model. This is a good model but I don’t see from your writing how this model informed your research.P: 3

In this study Stephen's resilience model was used which emphasized the importance of defining the concept of resilience as a process, also was proposed in nursing student setting. 

P: 5

Exploratory questions were asked following the main questions that were derived from the Stephen resilience model. The main questions included “explain about the situation experienced stress during nursing education.”, “which protective factors did you apply in these situations?”, and “explain about cumulative success do you have acquired?”. 

3-The title and introduction look as if you are developing an entirely new tool. However, the factor structures seem to suggest that you were validating the NSARI tool in the Iranian context. Can you please clarify this?P:5,6 

The data in the qualitative phase were collected through semi-structured interviews based on resilience Stephen's model in nursing students. Item generation and developing inventory were done based on the categories extracted from the qualitative phase. Then, psychometric properties were evaluated in the quantitative phase. …….

the first draft of the inventory is developed based on the findings of the directed content analysis.

4- The qualitative results did not indicate which items were generated and how they were elucidated from the qualitative study. It will be good to get a clear path on how items are generated from the interview data before psychometric analysis. At the moment, your qualitative results did not capture that. It is advisable to, at least, state the themes, categories, and sub-categories backed by participant quotes. This will provide evidence on how you arrived at the items that are used in the quantitative phase. P:10

An example of how the data were analyzed is shown in Table1.

P: 9

These codes were allocated to three themes, 9 categories (perceived stress: environment, relationship, Social-standing, Burnout, moral distress; protective factor: personal, social; cumulative success: Well-being, enhanced coping /adaptive abilities), and 31 subcategories based on interviews in the field of nursing student and Stephen’s model.

5- In line 132, you indicated the inventory was revised by 13 nursing students in the qualitative face validity stage. How were these revisions done because your qualitative methodology only talk about how the items would be developed, and not about revision. The psychometric properties (in quantitative phase) includes face validity, content validity, construct validity,and reliability. Face validity is assessed both qualitatively and quantitatively.In the qualitative face validity, participants are asked to read aloud the items, explain what each item means, and determine the problematic or ambiguous words for revision.Then items are revised and rewritten according to their views. Quantitative face validity is performed to determine the importance of each item. 

6- Under the results in the quantitative phase, Line 209-2016 seems to be talking about method rather than results. Can you move this to the methods section?This paragraph is related to the result of face and content validity. 

7- In line 209-210, you indicated that “In the psychometric evaluation phase, confusing items are revised and duplicate, redundant items are merged and essential items are added based on the results of the face and content validity” How were these revisions and merging done. Where from the ‘essential items”? 

Face validity, as we know is usually done by experts. Who were the ‘experts’ in your study? Please kindly explain how this was done as it forms an essential aspect of instrument development. P: 6

Face validity refers only to the appearance of the instrument to the responders; that is, an instrument appears to measure what the test constructor claims it measures and also motivate participants to respond the survey[16]. In terms of qualitative validity, participants were asked to read the items aloud, explain the meaning of each item, and identify problematic or ambiguous words for reconsideration Face validity is not validity in the true sense and refers only to the appearance of the instrument to the layperson; that is, if upon cursory inspection, an instrument appears to measure what

the test constructor claims it measures, it is said to have face validity. ) If an instrument has face validity, the layperson is more apt to be motivated to respond, thus its presence may serve as a factor in increasing response rate. Face validity,

when it is present, however, does not provide evidence for validity, that is, evidence that the instrument actually measures what it purports to measure). Carolyn Feher Waltz, Ora Lea Strickland, Elizabeth R. Lenz, Measurement in Nursing and Health Research. Fourth Edition(

8- I am not too conversant with your measurement of convergent and divergent validity. However, I do know that the common, and probably the best, way of determining divergent validity for instance is to measure the focal construct (in this case resilience) and another closely related concept using the same instrument. Divergent validity would be established when the two measures are weakly correlated. This is to ensure that the tool is not measuring other constructs. I know this may not have been your intent but it is worth knowing. The criterion of Fornell-Larcker was used to estimate the convergent and divergent validity based on the average variance extracted (AVE), the maximum variance (MSV).

(Fornell C, Larcker DF. Evaluating structural equation models with unobservable variables and measurement error. Journal of marketing research. 1981;18(1):39-50.

Factor Analysis and Discriminant Validity: A Brief Review of Some Practical Issues )

Reviewer4

. Perhaps creating a specific section for the detailed description of the participants, another session for the details of the procedure, another session for the measurements and finally statistical analysis, could improve the reading considerably.P:4 

As far as this study is a mixed method study with two phase qualitative and quantitative with their own pariticipants and procedures , therefore the main devision is based on these phases. 

For construct validity, the authors used a methodological approach that is commonly used in the field; however, the approach assumes a linear relationship between the items and the factor. Also, this methodology assumes a gaussian distribution of the outcome. This approach can be a problem for the construction of domains. Accurate methods for detecting relation between items are relevant for designing and determine the factor distribution. I suggest using Item Response Theory (IRT), specifically Graded Response Model (GRM; Samejima, 1969, 1997), Generalized Partial Credit Model (GPCM; Muraki, 1992) or multidimensional graded response model (MGRM). The IRT models examine the overall response patterns across all the items, while factor analytic methods examine covariances between the individual items. As a consequence of evaluating item response patterns, the parameter estimates provide insight into how the items work. Finally, the IRT estimates constructs assuming nonlinear relationship between latent traits and item responses.For IRT authors can use mirt package. It’s a powerful package and free tool for psychometric analyses: Chalmers, R. P. (2012). mirt: A multidimensional item response theory package for the R environment. Journal of Statistical Software, 48(6), 1-29.Although ther authors believe that test development decisions could be improved by using additional information about item responses, the purpose of this study was development and psychometric properties of instrument, therefore item selection was based on indices of difficulty and discrimination (classic approach). (Introduction to Classical and Modern Test Theory. Linda Crocker, James Aigina)

P: 19 ; Also, in the suggestion section it is suggested: 

To ensure that the instrument is psychometrically robust, its psychometric properties should be tested using different statistical methods and among different populations. Therefore, it is suggested that the Rasch model is used to test the NSARI psychometric properties of additional information.

Another suggested approach to construct validity is an exploratory structural equation modeling (see Asparouhov et al., 2009) approach.

Asparouhov, Tihomir, and Bengt Muthén. "Exploratory structural equation modeling." Structural equation modeling: a multidisciplinary journal 16.3 (2009): 397-438.

I’m agreed with authors, it’s necessary to estimate age and sex effect. However, I suggest reviewing the papers related to this issue of Rivera and Wim Van der Elst. The study was a mixed method study that represents an alternative methodological approach to traditional qualitative or quantitative research approaches to develope and psychometric properties of resilienceinstrument, the main purpose was not development of a resilience model, therefore item selection was based on indices of difficulty and discrimination (classic approach).

p:20

In addition the amount of data in the mixed method study is high, due to the word restriction in the article I would suggest in the suggestion section that it is consider in further study. 

Please add the exclusion criteria that were used to enroll participants in the study. P:4

The study population includes all bachelor's nursing students who intended to participate in this study. The exclusion criteria was any circumstance that may interfere with study participation

Please provide information about how the sample size was calculated in each group. P: 5; In qualitative research there is not exact sample size and data saturation is importan.

To achieve maximum data diversity students were selected of both gender, different semesters, private and public universities. The interviews were perpetuated until data saturation.

For instruments with less than forty items, samples of 200 were reported to be adequate.( Lin SY, Tseng WT, Hsu MJ, Chiang HY, Tseng HC (2017) A psychometric evaluation of the Chinese version of the nursing home survey on patient safety culture. Journal of clinical nursing 26: 4664-4674.)

Provide more information about face and content validity data analyses technique. P:6 

The psychometric properties (in quantitative phase) includes face validity, content validity, construct validity,and reliability. Face validity is assessed both qualitatively and quantitatively.In the qualitative face validity, participants are asked to read aloud the items, explain what each item means, and determine the problematic or ambiguous words for revision.Then items are revised and rewritten according to their views. Quantitative face validity is performed to determine the importance of each item. 

Face validity is not validity in the true sense and refers only to the appearance of the instrument to the layperson; that is, if upon cursory inspection, an instrument appears to measure what the test constructor claims it measures, it is said

to have face validity. ) If an instrument has face validity, the layperson is more apt to be motivated to respond, thus its presence may serve as a factor in increasing response rate. Face validity, when it is present, however, does not provide

evidence for validity, that is, evidence that the instrument actually measures what it purports to measure). Carolyn Feher Waltz, Ora Lea Strickland, Elizabeth R. Lenz, Measurement in Nursing and Health Research. Fourth Edition(

p:7 

Validity has then been commonly separated into various types—such as construct validity, content validity, onvergent and divergent validity, face validity, and factorial validity. COSMIN defi ned content validity as “the degree to which the content of an instrument is an adequate refl ection of the construct to be measured”.

---

## [Editor Report · Decision Letter 1]

12 Feb 2021

PONE-D-20-37513R1

Development and psychometric properties of the Nursing Student Academic Resilience Inventory (NSARI): A Mixed-Method study

PLOS ONE

Dear Dr. ghods,

Thank you for submitting your manuscript to PLOS ONE. After careful consideration, we feel that it has merit but does not fully meet PLOS ONE’s publication criteria as it currently stands. Therefore, we invite you to submit a revised version of the manuscript that addresses the points raised during the review process.

The study behind the manuscript has some merits and some limitations; however, the manuscript in its revised form is not still able to clearly underline them. The Authors answered only partially to most of the Reviewers’ concerns; very often they simply reported, in response, exactly what they had already written in the manuscript without any change or explanation when they did not agree. I want to emphasize here that responding appropriately, comprehensively, and convincingly to Reviewers is mandatory for a manuscript to be accepted for publication.

Based on my consideration of some merits of the research, I ask the Authors to further revise their manuscript following my advises listed below as Additional Editor Comments.

We look forward to receiving your revised manuscript.

Kind regards,

Prof. Paola Gremigni, Ph.D.

Academic Editor

PLOS ONE

**Additional Editor Comments:**

1) Introduction. Response to Reviewer #2-second point was not exhaustive. You can cite and synthetize concepts expressed by the review of resilience measures by Windle, Bennet, Noyes (2011 in HQoLO) to justify the necessity of a new scale like your.

2) The issue of missing data raised by Reviewer #2 is of maximum importance in research. It is not a matter of only answering the Reviewer, but to add an explanation in the manuscript for the readers. So, you should report your response statement into the manuscript when you described the methods.

3) As Reviewer #2 observed, a robust description of the factors that emerged is missing in the results section. Your answer to this observation was not convincing; thus, I ask you to move from the Discussion to the Results the definitions of the constructs measured with the 6 factors. Particularly: lines 315-317 for optimism; add a definition of effective communication that is missing; line 334 for self-esteem; line 339 for self-awareness; add a definition for trustworthiness that is missing everywhere; lines 357-361 for self-regulation.

4) The last concern by Reviewer #2 could be answered more effectively. I would like to see a new paragraph named “Study design and Procedures” at the beginning of the Methods. There, you should state that you used an exploratory sequential mixed-method approach, citing a recent reference to this approach (e.g, Creswell and Plano Clark, 2011 or Onwuegbuzie and Combs 2010 or others) and explaining that this approach draws on the strengths of both the qualitative and quantitative methodologies. (This is, indeed, a strength of your study). Followed by lines 103-107.

5) Reviewer’s #3 point 2 should be thoroughly answered. The phrase in lines 75-77 is grammatically incorrect. You can say that in this study, Stephen's resilience model of resilience in nursing students was used. This model emphasizes the importance of defining the concept of resilience as a process based on antecedent/perceived adversities, attributes/protective factors, and consequences/cumulative successes. The proposed model combines perceived adversities with the use of individual protective factors to effectively cope and/or adapt (page 130, Stephens, 2013).

6) To clarify any doubt like Reviewer’s #3 point 3, you should add the word “new” in the Abstract, line 33 “…to develop and validate a theory-driven new inventory labeled …”.and in the Introduction, line 94 “…..to develop and design a new valid and reliable inventory….”

7) Reviewer‘s #3-point 6 request should be accomplished. Lines 229-230 describe the method of the quantitative phase; thus, it is better to move it above after line 135.

8) To come to a compromise between the first request by Reviewer # 4 and your position, I ask you to eliminate the too generic Participants paragraph. Data collection of the qualitative phase should also include indication about recruitment of the participants and a more detailed description of the sample. For example: thirteen bachelor’s nursing students aged 18-25 years (indicate the percentage of gender), one (female?) nurse, and one (male?) trainer were recruited by a convenient sampling approach and participated only in the qualitative phase of the study (if it is true)

9) In the Quantitative phase a Data collection paragraph is totally missing. There, you should report how many independent samples you recruited for this phase and how. For example, suddenly 36 students (line 155) appeared that you had never talked about before. Again 11 experts (line 234) appeared by surprise when you reported the results of content analysis, too late to mention this small sample. Who they were? Experts of what? How and where have been recruited?

You should summarize briefly in this new Data collection paragraph the samples issues: how many samples did you involve in the quantitative phase of the study and where and how they were recruited? For example: you recruited (how?) 36 students for running the calculations for quantitative face validity, item analysis, test for normality and outliers, and test-retest reliability, or not?. You recruited 11 experts (who, where, how?) to calculate content validity indices. You recruited 200 students (where and how?) to run EFA and other 205 students (where and how?) to run the subsequent CFA” Or you calculated all the preliminary quantitative analyses using the answer of the first 13 students? In this case, the power of these calculations could be very low….. I am just guessing because it is not at all clear. Then, you will describe better the characteristics of those samples in the appropriate subsequent paragraphs.

10) The phrase “All the statistical analyses were calculated by SPSS-AMOS25 and SPSS R-Menu v2.0.” should be moved at the end of the quantitative phase, following line 204.

11) Ethical consideration: it is not enough to explain to the participants the volunteering and confidentiality of data, you should also explain the objectives of the study and collect their informed consent to participate in the study.

12) Results line 241: “Finally 13 items were eliminated from the analysis.” You should explain why, based on what criteria?

13) Construct validity line 243: you should report the sample size of the sample used here and percentage of female, besides absolute number.

14) Confirmatory factor analysis line 254. As in the previous paragraph, you should report here the sample size for this part, age and gender.

15) ANOVA results line 290. Why did you not calculate multiple comparisons between subgroups based on semester?

16) Scoring items line 294: you should explain why you gave an indication for obtaining a total (summative) score instead of 6 factors scores, especially because in the Discussion you dedicated a large room to the description of the six constructs.

17) Discussion: you cannot say “The CFA results detected the fitness of the final model”, because a number of fit indices you reported were under the recommended cutoff. Thus, you should underline and comment it.

18) Limitations should be expanded: 1) a probably sample disproportion regarding gender; 2) the use of validity indices based only on the internal structure of the scale. In future studies other methods could be used, like the heterotrait-monotrait (HTMT) ratio rates correlations, or the multitrait–multimethod matrix (MTMM), also correlating the instrument with external comparators (e.g, other measures of resilience); 3) the use of modification indices for fitting CFA results to your own data. This is an object of much disagreement among quantitative methodologists since this is a kind of “post hoc model modification” or a data-driven modification of the original hypothesized model. Other approaches could be used in future study to avoid this issue, like the exploratory structural equation modeling (ESEM) (see Asparouhov et al., 2009) as suggested by Reviewer #4.

Minor issues

A linguistic revision is still needed; a few present tense is still there, and many phrases are not understandable.

---

## [Author Response · Author response to Decision Letter 1]

22 Feb 2021

Dear Editor-in-Chief of PLOS ONE Journal

Dr. Joerg Heber

Response to reviewers

Thank you for your thorough review and consideration of our manuscript “Development and psychometric properties of the Nursing Student Academic resilience Inventory (NSARI): A Mixed-Method study" that was submitted to PLOS ONE Journal.

We believe that the comments and suggestions that were recommended by the reviewers have informed a much improved and more fully developed paper that will offer an important contribution to the field.

Given the word length requirements for clinical research briefs, we are somewhat stymied by the extent of details we can provide, However, and of critical importance, attention has been paid to developing a clearer rationale that addresses the significance of this work. We have carefully reviewed the comments and have revised the manuscript accordingly. Our responses are given in a point-by-point manner below. Please find the revised manuscript that is edited according to reviewers’ and editors’ comments, and all these changes are highlighted.

We hope the revised version is now suitable for publication and look forward to hearing from you in due course. 

Please feel free to contact me with any questions and concerns. I look forward to hearing from you regarding the manuscript.

Thanks for your kind attention to the manuscript.

Sincerely, Corresponding author 

Reviewer 1 Response

1. It is better to write Sample size same in the abstract (15) and the main study (13).

 P: 2 

Due to this study is a mixed method study and has qualitative and quantitative phase , also in quantitative phase has several stage so the number of participants is different in every stage but overall, the number of participants in the qualitative stage was 15 participant including 13 nursing students, one nurse and one trainer.

P:5

thirteen bachelor’s nursing students aged between 18 and 25 years old (5 women and 8 men), one female nurse, and one female trainer who had good communication skills and intended to participate in this study, were recruited using purposive sampling approach who have then participated only in the qualitative phase of the study.

2. If you write the Abbreviations (minimum) in the study readers can read easily your article. P:8,9,10

The abrevation are written, and highlited. 

3. You write the conclusion section in the abstract. It is better to write in the main study although it is optional (journal requirement) The abstract was written based on instruction for author.

4.Exploratory factor of the NSARI seems very good. Thank you for your kind comment.

5. in Table 2 better to write abbreviations below the table. P: 12.

It is written , and highlighted

Reviewer2 Response

Grammar, word choice & sentence structure issues are barriers to fully comprehending the manuscript content. Most of the manuscript is also written in present/future tense; if the studies are completed, it should all be written in past tense. The manuscript is edited by native English edit and the certificate letter is attached. 

There is not enough detail about current instruments and what makes this instrument necessary. P: 4

[6]. However, different studies have found different factor structures of resilience tools in different populations and cultures [7, 8]. Accordingly, it may be due to the reason that despite the prevalence of stress and the obligation to design an appropriate tool to assess students' resilience, it is not yet clear which factors measure the resilience in nursing students. However, it was shown that having an understanding of resilient characteristics and the protective factors that promote resilience in nursing student can enable them to cope with adversity during their education period [9].

These diverse approaches to measure resilience affected by different natures of potential risk factors and protective processes have led to some contradictions [10]. Due to establishing the resilience concept as a meaningful concept in research and clinical practice, it is inevitable to determine its distinctive factors and measure its factors using a valid and reliable method [11]. Notably, the evaluation of designed interventions to promote resilience demands reliable and valid measures[10].

In this way, researchers and clinicians do not have strong evidences regarding the choice of resilience measurement tools, which may make inappropriate choices in the field of study [12]. However, measuring resilience has been done by a number of researchers such as Charney[13], Wagnild[14], Kobasa[15]. Additionally, the mean score of resilience in these studies varies from 83 to 104[16, 17] because of using different generic scales. As a result, this creates difficulties in comparing mean score across studies, even if the nursing student experience similar adversities. In order to address these concerns, this study aimed to develop and design a new valid and reliable inventory for measuring resilience in nursing students. 

P. 7 lines 150-151: the authors state “There is no missing data in the study due to completing the surveys by an online Survey software (Porsline).” The reviewer does not understand how having surveys completed online assures no missing data P: 11

There was no missing data in this study due to completing the surveys by Porsline. In this online survey software, responders must respond to all questions because following and responding one question depends on answering the previous one. 

A description of the factors comes in the discussion section, but there is lacking a robust description of these in the results section P:13,14

Correspondingly, the first factor was optimism, which can be defined as people's positive expectations of what is happening in their lives [43]. In addition, having a positive optimistic attitude can help in more effectively dealing with stress [44]. The second factor was labeled as communication defined as the exchange of information, ideas, and feelings among people who use speech or other means[45]. Self-esteem/ evaluation as the third factor is known as a prime predictor of stress management indicateingthe degree of belief in the ability, importance, success, and individual’s competency [46]. The fourth factor was identified as self-awareness. Accordingly, this psychological component implies awareness of feelings, motivations, self-concept, and personality [47]. Trustworthiness as a worth demanding nurses to foster certain character attitudes and to strength effective task performance [48], was characterized as the fifth factor. Finally, self-regulation as the sixth factor was defined as the student’s attempt to manage learning processes focused on fulfilling goals [49].

The reviewer believes that content in this manuscript was ambitious and likely too much to include in a single manuscript. Ultimately, this detracted significantly from fully comprehending and appreciating the processes of instrument development and the value of this newly developed instrument. P: 5

Methods

Study design and Procedures

This exploratory sequential mixed-method study[18] was conducted in Iran from May 2019 to August 2020. It is assumed that all methods in this regard have bias and weaknesses, but based on triangulating data sources in this kind of design as well as the collection of both quantitative and qualitative data, the weaknesses of each form of data is neutralized[18]. The data in the qualitative phase were collected through performing semi-structured interviews based on the resilience Stephen's model in nursing students. Item generation and developing inventory were also done based on the categories extracted from the qualitative phase. Finally, psychometric properties were evaluated in the quantitative phase. 

Qualitative phase:

Data collection

Data analysis

Quantitative phase:

Data collection

Data analysis

Reviewer3 Response

1- kindly check the grammar and ensure correct sentence structures

 The manuscript is edited by native English edit and the certificate letter is attached.

2- You indicated using Stephen's Resilience Model. This is a good model but I don’t see from your writing how this model informed your research. P: 3,4

In this study, in line with different resilience theories, Stephen's resilience model of resilience in nursing students was used. Accordingly, this model emphasizes on the importance of defining the concept of resilience as a process performed based on antecedent/perceived adversities, attributes/protective factors, and consequences/cumulative successes. Moreover, the proposed model combines perceived adversities with the use of individual protective factors to cope and/or be adapted with the difficulties effectively. The cumulative successes of these events would lead to the increased resilience demonstrated by the enhanced coping/adaptive abilities and well-being status” [1].

P: 5,6

The data in the qualitative phase were collected through performing semi-structured interviews based on the resilience Stephen's model in nursing students.

Exploratory questions were asked following asking the main questions derived from the Stephen resilience model. The main questions included “explain about the situation experienced stress during nursing education.”, “which protective factors do you apply in these situations?”, and “explain about cumulative success that you have acquired”. 

3-The title and introduction look as if you are developing an entirely new tool. However, the factor structures seem to suggest that you were validating the NSARI tool in the Iranian context. Can you please clarify this? P: 2 

This study aimed to develop and validate a new inventory theory-driven labeled Nursing Student Academic Resilience Inventory.

P: 5

this study aimed to develop and design a new valid and reliable inventory for measuring resilience in nursing students. 

P:5,6 

The data in the qualitative phase were collected through performing semi-structured interviews based on the resilience Stephen's model in nursing students. Item generation and developing inventory were also done based on the categories extracted from the qualitative phase.

P:7

Data from the qualitative phase were used to develop Nursing Student Academic Resilience Inventory (NSARI) for the quantitative phase. 

4- The qualitative results did not indicate which items were generated and how they were elucidated from the qualitative study. It will be good to get a clear path on how items are generated from the interview data before psychometric analysis. At the moment, your qualitative results did not capture that. It is advisable to, at least, state the themes, categories, and sub-categories backed by participant quotes. This will provide evidence on how you arrived at the items that are used in the quantitative phase

 P:12

An example of how the data were analyzed is shown in Table1.

P: 11,12

These codes were allocated to three themes, 9 categories (perceived stress: environment, relationship, Social-standing, Burnout, moral distress; protective factor: personal, social; cumulative success: Well-being, enhanced coping /adaptive abilities), and 31 subcategories based on interviews in the field of nursing student and Stephen’s model.

5- In line 132, you indicated the inventory was revised by 13 nursing students in the qualitative face validity stage. How were these revisions done because your qualitative methodology only talk about how the items would be developed, and not about revision P: 7

Face validity only refers to the appearance of the instrument to the responders; in other words, an instrument appears to measure what the test constructor intended to measure and also motivates the participants to respond to the survey[21]. Confusing items were revised, duplicate, redundant items were merged, and essential items were added based on the results of the face and content validities. In terms of qualitative face validity, 13 nursing students were requested through convenience sampling to read the items loudly, explain the meaning of each item, and then identify problematic or ambiguous words for reconsideration.

6- Under the results in the quantitative phase, Line 209-2016 seems to be talking about method rather than results. Can you move this to the methods section?

 It is moved to the method section.

7- In line 209-210, you indicated that “In the psychometric evaluation phase, confusing items are revised and duplicate, redundant items are merged and essential items are added based on the results of the face and content validity” How were these revisions and merging done. Where from the ‘essential items”? 

Face validity, as we know is usually done by experts. Who were the ‘experts’ in your study? Please kindly explain how this was done as it forms an essential aspect of instrument development P: 7

Face validity only refers to the appearance of the instrument to the responders; in other words, an instrument appears to measure what the test constructor intended to measure and also motivates the participants to respond to the survey[21]. Confusing items were revised, duplicate, redundant items were merged, and essential items were added based on the results of the face and content validities. In terms of qualitative face validity, 13 nursing students were requested through convenience sampling to read the items loudly, explain the meaning of each item, and then identify problematic or ambiguous words for reconsideration. 

 Face validity is not validity in the true sense and refers only to the appearance of the instrument to the layperson; that is, if upon cursory inspection, an instrument appears to measure what the test constructor claims it measures, it is said to have face validity. ) If an instrument has face validity, the layperson is more apt to be motivated to respond, thus its presence may serve as a factor in increasing response rate. Face validity,when it is present, however, does not provide evidence for validity, that is, evidence that the instrument actually measures what it purports to measure). Carolyn Feher Waltz, Ora Lea Strickland, Elizabeth R. Lenz, Measurement in Nursing and Health Research. Fourth Edition(

8- I am not too conversant with your measurement of convergent and divergent validity. However, I do know that the common, and probably the best, way of determining divergent validity for instance is to measure the focal construct (in this case resilience) and another closely related concept using the same instrument. Divergent validity would be established when the two measures are weakly correlated. This is to ensure that the tool is not measuring other constructs. I know this may not have been your intent but it is worth knowing P:9

The criterion of Fornell-Larcker was used to estimate the convergent and divergent validity based on the average variance extracted (AVE), the maximum variance (MSV).

(Fornell C, Larcker DF. Evaluating structural equation models with unobservable variables and measurement error. Journal of marketing research. 1981;18(1):39-50.).

Factor Analysis and Discriminant Validity: A Brief Review of Some Practical Issues )

P: 9,16

Additionally, Heterotrait-monotrait ratio of correlations (HTMT) criterion was clculated. It was established that all values in the HTMT matrix table should be less than .85[32]. 

 (Table 4).

Reviewer4 Response

. Perhaps creating a specific section for the detailed description of the participants, another session for the details of the procedure, another session for the measurements and finally statistical analysis, could improve the reading considerably. P: 5

Methods

Study design and Procedures

This exploratory sequential mixed-method study[18] was conducted in Iran from May 2019 to August 2020. It is assumed that all methods in this regard have bias and weaknesses, but based on triangulating data sources in this kind of design as well as the collection of both quantitative and qualitative data, the weaknesses of each form of data is neutralized[18]. The data in the qualitative phase were collected through performing semi-structured interviews based on the resilience Stephen's model in nursing students. Item generation and developing inventory were also done based on the categories extracted from the qualitative phase. Finally, psychometric properties were evaluated in the quantitative phase. 

Qualitative phase:

Data collection

Data analysis

Quantitative phase:

Data collection

Data analysis

For construct validity, the authors used a methodological approach that is commonly used in the field; however, the approach assumes a linear relationship between the items and the factor. Also, this methodology assumes a gaussian distribution of the outcome. This approach can be a problem for the construction of domains. Accurate methods for detecting relation between items are relevant for designing and determine the factor distribution. I suggest using Item Response Theory (IRT), specifically Graded Response Model (GRM; Samejima, 1969, 1997), Generalized Partial Credit Model (GPCM; Muraki, 1992) or multidimensional graded response model (MGRM). The IRT models examine the overall response patterns across all the items, while factor analytic methods examine covariances between the individual items. As a consequence of evaluating item response patterns, the parameter estimates provide insight into how the items work. Finally, the IRT estimates constructs assuming nonlinear relationship between latent traits and item responses.

For IRT authors can use mirt package. It’s a powerful package and free tool for psychometric analyses: Chalmers, R. P. (2012). mirt: A multidimensional item response theory package for the R environment. Journal of Statistical Software, 48(6), 1-29. Although ther authors believe that test development decisions could be improved by using additional information about item responses, the purpose of this study was development and psychometric properties of instrument, therefore item selection was based on indices of difficulty and discrimination (classic approach). (Introduction to Classical and Modern Test Theory. Linda Crocker, James Aigina)

P: 22, 

To ensure that the instrument is psychometrically robust, its psychometric properties should be tested using different statistical methods such as the Rasch model and structural equation modeling, in order to estimate age and gender effects.

Another suggested approach to construct validity is an exploratory structural equation modeling (see Asparouhov et al., 2009) approach.

Asparouhov, Tihomir, and Bengt Muthén. "Exploratory structural equation modeling." Structural equation modeling: a multidisciplinary journal 16.3 (2009): 397-438.

I’m agreed with authors, it’s necessary to estimate age and sex effect. However, I suggest reviewing the papers related to this issue of Rivera and Wim Van der Elst The study was a mixed method study that represents an alternative methodological approach to traditional qualitative or quantitative research approaches to develope and psychometric properties of resilienceinstrument, the main purpose was not development of a resilience model, therefore item selection was based on indices of difficulty and discrimination (classic approach).

P:22

it is suggested in limitation section. 

Please add the exclusion criteria that were used to enroll participants in the study P:5

The study population includes all bachelor's nursing students who intended to participate in this study. The exclusion criteria was any circumstance that may interfere with study participation

Please provide information about how the sample size was calculated in each group Due to a mixed method study has a multiple phase and in different phase there is different numbers of participant so in every stage the number of participants and the way of selection is written.

P: 6

Qualitative phase

 In qualitative research there is not exact sample size and data saturation is important. 

To achieve maximum data diversity students were selected of both gender, different semesters, private and public universities. In qualitative studies, sample size cannot be pre-established, so much so the interviews were perpetuated until data saturation[Morse, 2002 #1949].

P: 7,8

Quantitative phase

Data collection

Data from the qualitative phase were used to develop Nursing Student Academic Resilience Inventory (NSARI) for the quantitative phase. Face validity was tested by 13 nursing students. Moreover, 11 experts in instrument development, nursing, and psychology were invited by email to validate this inventory in terms of content validity. All these experts were professors from the college of nursing and psychology and expertise in tool’s validation. Based on the Polit et al.’s study (2015), the size of the expert panel usually is between 3 and 12. For construct validity, those studies with the sample size of 100 or more were considered as “excellent”. Of note, the sample in CFA should be different from the sample used to finalize the instrument using EFA. According to the rule of thumb, a minimum of 200 cases in the analyses was considered to be enough. [19]. Finally, 200 nursing students for EFA as well as 205 nursing students for CFA analysis were recruited in this study.

Face and Content validity

In terms of qualitative face validity, 13 nursing students through convenience sampling were requested to….

CVR and CVI Were done by 11 experts in instrument development, nursing, and psychology to determined

Item Analysis 

36 nursing students were selected based on stratified sampling to recognize…

Exploratory Factor Analysis

in a cross‑sectional study, 200 nursing students of private and public universities from 6 different provinces in Iran participated in the study on account of convenience sampling.

Confirmatory Factor Analysis

Confirmatory factor analysis was done on a different sample including 205 nursing students who were selected by convenience sampling to…..

For instruments with less than forty items, samples of 200 were reported to be adequate.( Lin SY, Tseng WT, Hsu MJ, Chiang HY, Tseng HC (2017) A psychometric evaluation of the Chinese version of the nursing home survey on patient safety culture. Journal of clinical nursing 26: 4664-4674.)

P: 9

The Kaiser-Meyer-Olkin (KMO) statistic of sampling adequacy and the Bartlett test of sphericity were calculated to check the suitability of the data for factor analysis.

Provide more information about face and content validity data analyses technique P:7,8

Face and Content validity

Face validity only refers to the appearance of the instrument to the responders; in other words, an instrument appears to measure what the test constructor intended to measure and also motivates the participants to respond to the survey[21]. Confusing items were revised, duplicate, redundant items were merged, and essential items were added based on the results of the face and content validities. In terms of qualitative face validity, 13 nursing students were requested through convenience sampling to read the items loudly, explain the meaning of each item, and then identify problematic or ambiguous words for reconsideration. Following revising the items of the designed inventory according to the comments of the participants, at the qualitative face validity stage, quantitative face validity was assessed using the impact factor method based on the following equation…

Due to the COSMIN definition, content validity is “the degree to which the content of an instrument is an adequate reflection of the construct to be measured”[19]. In this way, content validity ratio (CVR) was done by 11 experts to determine the essential items according to the cut-off point proposed in a study by Lawshe [23].

Additional Editor Comments

Introduction. Response to Reviewer #2-second point was not exhaustive P:4

It is clarified.

The issue of missing data raised by Reviewer #2 is of maximum importance in research. . So, you should report your response statement into the manuscript when you described the methods.

 P:11

It is written 

As Reviewer #2 observed, a robust description of the factors that emerged is missing in the results section. P: 13,14

It is explained 

I would like to see a new paragraph named “Study design and Procedures” at the beginning of the Methods P: 5

It is written.

Reviewer’s #3 point 2 should be thoroughly answered P: 3,4

 It is explained in more detailed.

To clarify any doubt like Reviewer’s #3 point 3, you should add the word “new. P:2,5

It is written.

Reviewer‘s #3-point 6 request should be accomplished. Lines 229-230 describe the method of the quantitative phase; thus, it is better to move it above after line 135. P: 7

It is moved

I ask you to eliminate the too generic Participants paragraph. Data collection of the qualitative phase should also include indication about recruitment of the participants and a more detailed description of the sample. P:5,6

It is written

In the Quantitative phase a Data collection paragraph is totally missing P;6,7

The phrase “All the statistical analyses were calculated by SPSS-AMOS25 and SPSS R-Menu v2.0.” should be moved at the end of the quantitative phase, following line 204. P: 10

Ethical consideration: it is not enough to explain to the participants the volunteering and confidentiality of data, you should also explain the objectives of the study and collect their informed consent to participate in the study. P:11

 It is explained.

Results line 241: “Finally 13 items were eliminated from the analysis.” You should explain why, based on what criteria? P:13

It explained.

Construct validity line 243: you should report the sample size of the sample used here and percentage of female, besides absolute number.

Confirmatory factor analysis line 254. As in the previous paragraph, you should report here the sample size for this part, age and gender. P:13

 It is written.

ANOVA results line 290. Why did you not calculate multiple comparisons between subgroups based on semester?

 P:17

 It is done and reported.

Scoring items line 294: you should explain why you gave an indication for obtaining a total (summative) score instead of 6 factors scores, especially because in the Discussion you dedicated a large room to the description of the six constructs. Composite measures often involve summing the item values to yield a total score, although sometimes, instrument developers recommend that the total score be the average across items so that the total score is on the same scale as the items. 

(Measurement and the Measurement of Change. Polit (2015))

Discussion: you cannot say “The CFA results detected the fitness of the final model”, because a number of fit indices you reported were under the recommended cutoff. Thus, you should underline and comment it.

 P: 22

It is declared in limitation.

Limitations should be expanded: a probably sample disproportion regarding gender;

 2) the use of validity indices based only on the internal structure of the scale. In future studies other methods could be used, like the heterotrait-monotrait (HTMT) ratio rates correlations, or the multitrait–multimethod matrix (MTMM), also correlating the instrument with external comparators (e.g, other measures of resilience); 

3) the use of modification indices for fitting CFA results to your own data. This is an object of much disagreement among quantitative methodologists since this is a kind of “post hoc model modification” or a data-driven modification of the original hypothesized model. Other approaches could be used in future study to avoid this issue, like the exploratory structural equation modeling (ESEM) (see Asparouhov et al., 2009) as suggested by Reviewer #4. P:22

the sample disproportion regarding gender; however, this is common because Nursing is a female-dominated profession [66]

P: 9, 16

2--HTMT is calculated

3- p:9, 21

It is discussed in discussion section. 

A linguistic revision is still needed; a few present tense is still there, and many phrases are not understandable. The manuscript is edited by native English edit and the certificate letter is attached.

---

## [Decision Letter · Decision Letter 2]

19 Apr 2021

PONE-D-20-37513R2

Development and psychometric properties of the Nursing Student Academic Resilience Inventory (NSARI): A Mixed-Method study

PLOS ONE

Dear Dr. ghods,

Thank you for submitting your manuscript to PLOS ONE. After careful consideration, we feel that it has merit but does not fully meet PLOS ONE’s publication criteria as it currently stands. In this second round of revision reviewers are not satisfied with the author's response to their concerns. Therefore, we invite you to submit a revised version of the manuscript that addresses the points raised during the review process.

We look forward to receiving your revised manuscript.

Kind regards,

Prof. Paola Gremigni, Ph.D.

Academic Editor

PLOS ONE

Reviewers' comments:

Reviewer's Responses to Questions

**Comments to the Author**

1. If the authors have adequately addressed your comments raised in a previous round of review and you feel that this manuscript is now acceptable for publication, you may indicate that here to bypass the “Comments to the Author” section, enter your conflict of interest statement in the “Confidential to Editor” section, and submit your "Accept" recommendation.

Reviewer #2: (No Response)

Reviewer #5: All comments have been addressed

Reviewer #6: (No Response)

Reviewer #7: (No Response)

Reviewer #8: (No Response)

2. Is the manuscript technically sound, and do the data support the conclusions?

Reviewer #2: Yes

Reviewer #5: Yes

Reviewer #6: No

Reviewer #7: Partly

Reviewer #8: Yes

3. Has the statistical analysis been performed appropriately and rigorously? 

Reviewer #2: Yes

Reviewer #5: I Don't Know

Reviewer #6: I Don't Know

Reviewer #7: I Don't Know

Reviewer #8: Yes

4. Have the authors made all data underlying the findings in their manuscript fully available?

Reviewer #2: Yes

Reviewer #5: Yes

Reviewer #6: Yes

Reviewer #7: Yes

Reviewer #8: Yes

5. Is the manuscript presented in an intelligible fashion and written in standard English?

Reviewer #2: No

Reviewer #5: Yes

Reviewer #6: No

Reviewer #7: No

Reviewer #8: Yes

6. Review Comments to the Author

Reviewer #2: The reviewer maintains that the topic is an important one; and that the instrument at face value has promise. While there is greater discussion about existing gaps in current instruments that this instrument addresses, this information could be presented more concisely. Developing the instrument based on a model strengthens validity, but what is lacking is a clear conceptualization of “resilience”. Describing the model components is not adequate as a definition of resilience. The reviewer believes this lack is at the core of the ambiguity found in the Introduction. Additionally, the manuscript appears to have been written in two different styles. The background and discussion appear to be much less clear and are in need of refinement. The methods section is much clearer and improved. Again, it is clear that the authors have done a significant amount of work. The manuscript, however, covers a large amount of information from multiple studies in the instrument development process, and the reviewer continues to wonder if this work would better be served by breaking it up into two (or more) manuscripts. Due to the significant amount of description, the extensive work is not appreciated as it could be if the manuscript covered less extensive, but more impactful content.

Reviewer #5: Please, note that I did not examine the statistical analysis. Please, consider statistical analysis elsewhere if not already undertaken.

Reviewer #6: 1. Generally, the paper contains too much textbook statements with reiterations. The paper needs to be organized in clear and parsimonious manner.

2. The paper needs a professional English editorial service.

3. Some statements are not supported with references as exampled, line 77-84. Line 84 has improper quotation mark at the end. Overall, the manuscript needs to be written in scientific writing format including table format.

4. If the aim of the paper is based on the premise of line 86-87 (different studies have found different factor structures of resilience tools in different populations and cultures), this paper seems to serve only for Iranian nursing students. When there are several studies on resilience from various populations and cultures, the next step for study is to try to develop the integrated tool of resilience, rather than to add one more study of a specific population or culture. Or the paper should address the robust rationale on why the development of the resilience specifically for Iranian nursing student population is important in addition to those of previous study samples. Then the findings should compare resilience too variations among various populations and cultures. In fact, there have been many studies on resilience in nursing so far.

5. line 94-96: Due to establishing the resilience concept as a meaningful concept in research and clinical practice, it is inevitable to determine its distinctive factors and measure its factors using a valid and reliable method [11]. It seems a little awkward sentence to me. The concepts of resilience have been established in many research and field.

6. The rationale for the study is not clear to me. On line 102-104: It is written, “Additionally, the mean score of resilience in these studies varies from 83 to 104[16, 17] because of using different generic scales. As a result, this creates difficulties in comparing mean score across studies, even if the nursing student experience similar adversities”. Then this study to develop and design a new valid and reliable inventory for measuring resilience in nursing students, seems to add one more different generic scale which may contribute to create difficulties in comparing mean across studies.

7. The criteria of subject seems still blurred. For example, on line 125, it is written ‘The exclusion criteria were any circumstance that may interfere with study participation.’ That is not the exclusion criteria. The circumstances of exclusion should be specified. As well on line 145, it is written, ‘the samples were selected with maximum variation.’ What does it mean by maximum variation?

8. For ethical considerations, it is extremely important to protect the right of study participants and to maintain their security when they are students and the researchers are professors at same institutions. The circumstances to obtain the consent of participation from them need to be more addressed in terms of the relationship between participants and researchers, who proposed them to participate into the study, the condition of free refusal, no influence to any school activities by the study, etc. The description on line 254-259 is too simple. As well the interview circumstances need to be explained more. The line 125 (interviews were conducted in a quiet room) is not enough.

9. Discussion needs to be focused the comparison of resilience tool among subjects in relation to Stephen’s resilience model based on the premise and aim of the study.

Reviewer #7: Feedback for the manuscript, entitled "Development and psychometric properties of the Nursing Student Academic Resilience Inventory (NSARI): A Mixed-Method study".

This study was focused on development of the nursing student academic resilience inventory (NSARI) and examination of psychometric properties of the NSARI using a combined approach of qualitative and quantitative methods. Overall, the writing of the manuscript is hard for me to follow. As mentioned in the reviews of the previous round, I feel the manuscript still need a further and careful proofread. There are typos and grammatical errors. Many places in the manuscript did not follow the APA. The APA. I would highly recommend the authors carefully re-reading the comments from the previous reviewers. The reviewers’ comments are extremely constructive but were ignored by the authors. Since there is still a need for a major revision, I will provide some broad suggestions listed below.

First, the author may want to develop a section called “Research Purpose and Specific Questions”. The authors only mentioned the purpose in one sentence on page 4. More detailed purpose can be addressed, and specific research questions can be developed. Specific research questions can guide the reader to better understand this study.

Second, the method section should be reorganized. As mentioned by a reviewer in the previous round, the participants section should provide more details regarding participants. Even though there are many phases with different participants, such as participants for interviews, EFA, and CFA. The same principle can be applied “Statistical Analyses”. Further, there are many statistics mentioned in the method, but I could not find any relevant results, such as Mahalanobis d-squared.

Third, in the results any statements should be along with supportive statistics. For instance, on page 11, the authors mentioned that “Finally, 13 items were eliminated from the analysis. Are there any statistics you should report to support your statistics?

Reviewer #8: My comments rotate mostly around Qualitative research and Ethics.

Line 111: I agree with a reviewer that mentioned that there is too much information in this manuscript, agree with their recommendation that the findings can be presented in two separate manuscript. Regarding the justification for combining, which is triangulation to reduce Bias, there are several ways triangulation in several ways. For example, I see in the qualitative methods, there was author triangulation. Very long manuscripts are not reader friendly.

Line 121:The author could try to maintain the language; use of man and woman vs male and female

Line 122:The authors mention that for qualitative Data collection, they considered those with good communication skills- Why, Could this bias the study by excluding the those who had no good communication skills. What is good communication skills- was it at the discretion of the researcher to define good communication skills. Possibility that people with good communication are characteristically different from those with poor communication skills and probably have different resilient mechanisms.

Line 123:The authors use the term intended to participate in the study. Not clear whether participants were consented to participate in the study. I like the fact the authors specify how the participants were selected. However wondering all people contacted participated in the study or there are some who could have decline and the possible reason if any decline

Line 125: What us any circumstances-Could may be cite an example of how any circumstance that may interfere with study participation may look like

Line 146:it is not clear to me If the researcher critically examined their own role,

potential bias and influence during (a) formulation of the research questions (b) data collection, including sample recruitment and choice of location

Line 256:Are you trying to describe verbal consent. other wise am worried that participants did not provide written informed consent. If writing, did it receive a waiver of consent from the REC?

Line 259:This statement is not clear. if you had talked about the consent process here; this would not be necessary;

Electronically, some authors will state that by clicking I agree to participate, is a taken as consent. But have to be given information about the study before accessing the survey

7. PLOS authors have the option to publish the peer review history of their article (what does this mean?). If published, this will include your full peer review and any attached files.

Reviewer #2: No

Reviewer #5: **Yes: **Adeniyi Olanrewaju Adeleye

Reviewer #6: No

Reviewer #7: No

Reviewer #8: **Yes: **Provia Ainembabazi

---

## [Author Response · Author response to Decision Letter 2]

30 Apr 2021

Dear Academic Editor

Prof. Paola Gremigni

Response to reviewers

Thank you for your thorough review and consideration of our manuscript “Development and psychometric properties of the Nursing Student Academic resilience Inventory (NSARI): A Mixed-Method study" that was submitted to PLOS ONE Journal.

We believe that the comments and suggestions that were recommended by the reviewers have informed a much improved and more fully developed paper that will offer an important contribution to the field.

We have carefully reviewed the comments and have revised the manuscript accordingly. Our responses are given in a point-by-point manner below. Please find the revised manuscript that is edited according to reviewers’ comments, and all these changes are highlighted, also the certification of Native English Edit company is uploaded. 

We hope the revised version is now suitable for publication and look forward to hearing from you in due course. 

Please feel free to contact me with any questions and concerns. I look forward to hearing from you regarding the manuscript.

Thanks for your kind attention to the manuscript.

Sincerely, Corresponding author

Reviewer #2:

What is lacking is a clear conceptualization of “resilience”. Describing the model components is not adequate as a definition of resilience. The reviewer believes this lack is at the core of the ambiguity found in the Introduction: P:3

Resilience evolved in the 1970s to understand why psychopathology was not always the consequences of risky environments in children [DeMichelis, 2016 #2004]. Resilience is described as a characteristic, process or outcome that depends on the theory accepted by the researcher[4]. Resilience’s theories can be outlined to three different models of resilience as “compensatory”, “protective”, or “inoculation/challenge” [DeMichelis, 2016 #2004]

The background and discussion appear to be much less clear and are in need of refinement: The background and discussion are refined

P: 3,4 (Introduction)

P: 19 (discussion)

The aim of present study was to develop a new inventory for measuring the resilience in nursing students. Although numerous tools have been designed in different contexts, there is only one study that was designed in nursing students. Yang et al study (2015) extracted 24 items through literature review and student interviews in two personal and contextual dimensions in Korean nursing. These dimensions included Self-confidence, Positivity, Coping ability, Emotional regulation, Structured style, Relationship, Social support (Yang YH, Kim EM, Yu M, Park S, Lee H. Development of the resilience scale for korean nursing college students. Korean Journal of Adult Nursing 2015; 27: 337-46.). As shown this study, resilience is multidimensional concept but some dimensions are the core of the resilience such as relationship and being able to cope. That was similar to the result of present study.

Reviewer #6:

1-the paper contains too much textbook statements with reiterationsbelow statements omitted and some statement paraphrases by Radan English Edit institute

P: 7 

an instrument appears to measure and also motivates the participants to respond to the survey

P:8 

Based on the Polit et al.’s study (2015), the size of the expert panel usually is between 3 and 12. For construct validity, those studies with the sample size of 100 or more were considered as “excellent”. Of note, the sample in CFA should be different from the sample used to finalize the instrument using EFA.

P:10

According to the COSMIN standards, besides validity and reliability, interpretability and scoring similarly are considered as important capabilities of a tool

2- The paper needs a professional English editorial service.:: The paper was edited again by www.NativeEnglishEdit.com

London

East End Road 27, N 3 3QT

United Kingdom

3-Some statements are not supported with references as exampled, line 77-84. 

Line 84 has improper quotation mark at the end: P: 3 

The paragraph mentioned to below reference that is cited: Stephens TM. Nursing student resilience: a concept clarification. Nursing forum. 2013;48(2):125-33.”

 The quotation mark is deleted

4- If the aim of the paper is based on the premise of line 86-87 (different studies have found different factor structures of resilience tools in different populations and cultures), this paper seems to serve only for Iranian nursing students. When there are several studies on resilience from various populations and cultures, the next step for study is to try to develop the integrated tool of resilience, rather than to add one more study of a specific population or culture. Or the paper should address the robust rationale on why the development of the resilience specifically for Iranian nursing student population is important in addition to those of previous study samples. Then the findings should compare resilience too variations among various populations and cultures. In fact, there have been many studies on resilience in nursing so far.: P: 3, 4

Previous studies conducted in Iran showed that nursing students are affected by various stressors… Resilience has been suggested as a beneficial solution for better dealing with challenges in nursing students.

The result of a review study yielded by Chmitorz et al. (2018) showed that various resilience scales have been developed [5]. Also, several studies have found different factor structures of resilience in different populations and cultures [6, 7]. In addition, there is only one exploratory study in nursing student setting[8]. Therefore, it is not yet clear which factors measure the resilience in nursing students [9] that seems to be related to different natures of potential risk factors and protective processes which have led to some contradictions while the evaluation of designed interventions to promote resilience demands reliable and valid measures. In this way, researchers and clinicians do not have strong evidences regarding the choice of resilience measurement tools, which may make inappropriate choices in the field of study [10].P: 19 

Although numerous tools have been designed in different contexts, there is only one study that was designed in nursing students. Yang et al study (2015) extracted 24 items through literature review and student interviews in two personal and contextual dimensions in Korean nursing. These dimensions included Self-confidence, Positivity, Coping ability, Emotional regulation, Structured style, Relationship, Social support [8]. As shown this study, resilience is multidimensional concept but some dimensions are the core of the resilience such as relationship and being able to cope. These results support the result of present study. Thomas et al (2016 ), also in integrative review declared factors that affect resilience in nursing students were grouped into three themes: support, time, and empowerment .

5- line 94-96: Due to establishing the resilience concept as a meaningful concept in research and clinical practice, it is inevitable to determine its distinctive factors and measure its factors using a valid and reliable method [11]. It seems a little awkward sentence to me. The concepts of resilience have been established in many research and field.: It is deleted. 

6- The rationale for the study is not clear to me. On line 102-104: It is written, “Additionally, the mean score of resilience in these studies varies from 83 to 104[16, 17] because of using different generic scales. As a result, this creates difficulties in comparing mean score across studies, even if the nursing student experience similar adversities”. Then this study to develop and design a new valid and reliable inventory for measuring resilience in nursing students, seems to add one more different generic scale which may contribute to create difficulties in comparing mean across studies.:P:4 

The result of a review study yielded by Chmitorz et al. (2018) showed that various resilience scales have been developed [6]. because resilience is a multidimensional construct that differes in different context, time, age, and life condition[Singh, 2010 #456]. several studies have found different factor structures of resilience in different populations and cultures [7, 8]. In addition, there is only one exploratory study in nursing student setting [Yang, 2015 #1793]. Therefore, it is not yet clear which factors measure the resilience in nursing students [9] that seems to be related to different natures of potential risk factors and protective processes which have led to some contradictions while the evaluation of designed interventions to promote resilience demands reliable and valid measures. In this way, researchers and clinicians do not have strong evidences regarding the choice of resilience measurement tools, which may make inappropriate choices in the field of study.

The paragraph about mean score is deleted.

7-The criteria of subject seems still blurred. For example, on line 125, it is written ‘The exclusion criteria were any circumstance that may interfere with study participation.’ That is not the exclusion criteria. The circumstances of exclusion should be specified.: P:6

 The exclusion criteria were any circumstance that may interfere with study participation such as speech and hearing impairment, having work experience in medical settings.

As well on line 145, it is written, ‘the samples were selected with maximum variation.’ What does it mean by maximum variation?P:7

To assure data credibility, the samples were selected with maximum variation (private and public universities, different semesters, age, sex, and score). Also, to cover the variation of the phenomenon, a small data batch analyzed carefully first and then determined what additional data will be needed ….

8- For ethical considerations, it is extremely important to protect the right of study participants and to maintain their security when they are students and the researchers are professors at same institutions. The circumstances to obtain the consent of participation from them need to be more addressed in terms of the relationship between participants and researchers, who proposed them to participate into the study, the condition of free refusal, no influence to any school activities by the study, etc. The description on line 254-259 is too simple: P: 11

Researchers were not professors at the same institutions. Therefore, there was any conflict of interest. The researcher referred to the education deputy and received the demographic list of students. In several sessions, students (in different educational semesters) invited to participate in the study. The aim of this study, data confidentiality, and voluntarily participation were explained. 

As well the interview circumstances need to be explained more. The line 125 (interviews were conducted in a quiet room) is not enough:P: 6

In the qualitative phase, the semi-structured individual interviews were conducted in a quiet room in the university or the educational class of hospital regarding to the student's choice. Students were selected through purposive sampling. The interviews started with asking an icebreaker question in which the individual is allowed to talk openly about the topic. Exploratory questions were asked following asking the main questions derived from the Stephen resilience model .

As well on line 145, it is written, ‘the samples were selected with maximum variation.’ What does it mean by maximum variation? 

P:7

the samples were selected with maximum variation (private and public universities, different semesters, age, sex, and score). Also, to cover the variation of the phenomenon, a small data batch analyzed carefully first and then determined what additional data will be needed

9- Discussion needs to be focused the comparison of resilience tool among subjects in relation to Stephen’s resilience model based on the premise and aim of the study: P: 16

Although numerous tools have been designed in different contexts, there is only one study that was designed in nursing students. Yang et al study (2015) extracted 24 items through literature review and student interviews in two personal and contextual dimensions in Korean nursing. These dimensions included Self-confidence, Positivity, Coping ability, Emotional regulation, Structured style, Relationship, Social support (Yang YH, Kim EM, Yu M, Park S, Lee H. Development of the resilience scale for korean nursing college students. Korean Journal of Adult Nursing 2015; 27: 337-46.). As shown this study, resilience is multidimensional concept but some dimensions are the core of the resilience such as relationship and being able to cope. That was similar to the result of present study

Reviewer #7

First, the author may want to develop a section called “Research Purpose and Specific Questions”. The authors only mentioned the purpose in one sentence on page 4. More detailed purpose can be addressed, and specific research questions can be developed. Specific research questions can guide the reader to better understand this study:p: 4

Research Purpose and Specific Questions

To address the explained concerns, and regard to lack of specific resilience tool in nursing students the aim of present study in the first phase was to examine the stress, protective factors and cumulative success that students have acquired. The overall aim of the study was to design a new valid and reliable inventory for measuring resilience in nursing students. To achieve this aim, there were two research questions;

(1) Which factors are considered to be related to the resilience of nursing students?

(2) Which of these factors has statistically significant effect on the resilience of nursing students?

Second, the method section should be reorganized. As mentioned by a reviewer in the previous round, the participants section should provide more details regarding participants. Even though there are many phases with different participants, such as participants for interviews, EFA, and CFA. The same principle can be applied “Statistical Analyses”. :It is reorganized overally.

P:5 

2.2 Participants 

In the qualitative phase, thirteen under graduated bachelor’s nursing students aged between 18 and 25 years old (5 female and 8 male), one female nurse, and one female trainer who had good communication skills and intended to participate in this study were recruited. Two students were dropout of keeping the education and one student was conditional student.

The quantitative phase included several stages. Face validity was tested by 13 nursing students (7female, 6 male) in different semesters. Moreover, 11 experts in instrument development, nursing, and psychology were invited by email to validate this inventory in terms of content validity. 200 nursing students for EFA as well as 205 nursing students for CFA analysis were recruited from 6 different provinces in Iran (age= 21.70 ±2.50; female=258, male= 147).

Further, there are many statistics mentioned in the method, but I could not find any relevant results, such as Mahalanobis d-squared.

P:7 

In the quantitative phase, the psychometric properties of the NSARI were assessed using face, content, and construct validities (exploratory and confirmatory factor analysis).

P:10 

The presence of a multivariate outlier was assessed by Mahalanobis d-squared (p< .001),.

The Mahalanobis distance is a well-known criterion which may be used for detecting outliers in multivariate data.( Appropriate Critical Values When Testing for a Single Multivariate Outlier by Using the Mahalanobis Distance. Kay I. Penny. Journal of the Royal Statistical Society. Series C (Applied Statistics) Vol. 45, No. 1 (1996), pp. 73-81 (9 pages) Published By: Wiley)

Third, in the results any statements should be along with supportive statistics. For instance, on page 11, the authors mentioned that “Finally, 13 items were eliminated from the analysis. Are there any statistics you should report to support your statistics? P:8

Using a pilot study, the internal consistency was assessed before the construct validity assessment to recognize potential problems in the NSARI through estimating Cronbach’s alpha and inter‑item correlation. The items whose corrected item-total correlation score was less than 0.3, were then removed from the analysis [18].

P: 13

13 items were eliminated from the analysis due to their corrected item-total correlation scores was less than 0.3 in Item analysis. 

Reviewer #8: 

Line 111: which is triangulation to reduce Bias, there are several ways triangulation in several ways. For example, I see in the qualitative methods, there was author triangulation: It is true that Validation of post positivist qualitative research relies upon using some strategies such as articulation of the research question., the use of more than one sources of data, providing a clear account of the process of data collection, negative case and triangulation of the findings. In this study all of them have done 

P:10.11

2.5 Trustworthiness in the mixed-method study

To develop Self-reflexivity, after transcribing each one of the interviews, the researchers analyzed the obtained data to uncover their preunderstandings. As well, the researchers consulted with team members on the extracted codes and themes and then described the data analysis process in detail and finally provided clear citations in this regard. 

To diminish the threats to the internal and external validities of the mix-method study, the participants with different experiences were selected. As well, none of the samples in the qualitative phase participated in the quantitative phase. Furthermore, designing the item pool was performed based on the main categories and sub-categories extracted in the qualitative phase. Consequently, all the stages of the study were carefully reviewed and verified by the other researchers [16, 36].

Line 121:The author could try to maintain the language; use of man and woman vs male and female:It is revised

Line 122:The authors mention that for qualitative Data collection, they considered those with good communication skills- Why, Could this bias the study by excluding the those who had no good communication skills. What is good communication skills- was it at the discretion of the researcher to define good communication skills. Possibility that people with good communication are characteristically different from those with poor communication skills and probably have different resilient mechanisms: Yes, it is true that people with good communication are characteristically different from those with poor communication skills and probably have different resilient mechanisms. Therefore, the researcher also interview with negative cases but the first interviewee was a female who was introduced by education deputy because she had good relationship with others and speak easily so we can reach a lot of data. 

P:5 - good communication skills including fluency in Persian language, no hearing or speech problems,

P: 5- Two students were dropout of keeping the education and one student was conditional student.

Line 123: The authors use the term intended to participate in the study. Not clear whether participants were consented to participate in the study. I like the fact the authors specify how the participants were selected. However wondering all people contacted participated in the study or there are some who could have decline and the possible reason if any decline: P: 11- The researcher referred to the education deputy and received the demographic list of students. In several sessions, students ( in different educational semesters) invited to participate in the study. The aim of this study, data confidentiality, and voluntarily participation were explained. In qualitative phase informed consent obtained was verbal due to in this way participants feel more intimate relationship for participating in interview sessions. the IRB at this site was inclined to this procedure.Thereafter, in the quantitative phase, surveys were completed through Porsline software.At the first of survey written informed consent was attached. The participants who did not want to participate were authorized not to complete the surveys.

Line 125: What us any circumstances-Could may be cite an example of how any circumstance that may interfere with study participation may look like: P: 6-The exclusion criteria were any circumstance that may interfere with study participation such as speech and hearing impairment, having work experience in medical settings.

Line 146:it is not clear to me If the researcher critically examined their own role,

potential bias and influence during (a) formulation of the research questions (b) data collection, including sample recruitment and choice of location: P:10, 11

Trustworthiness in the mixed-method study

To develop Self-reflexivity , after transcribing the each interview, the researchers analyses the data to uncover their preunderstandings, also researchers consulted with team members about the extracted codes and themes and described data analysis process in detail and provided clear citations. 

To diminish the threats to the internal and external validities of the mix-method study, the participants with different experiences were selected. As well, none of the samples in the qualitative phase participated in the quantitative phase. Furthermore, designing the item pool was performed based on the main categories and sub-categories extracted in the qualitative phase. Consequently, all the stages of the study were carefully reviewed and verified by the other researchers [15, 35].

Line 256:Are you trying to describe verbal consent. other wise am worried that participants did not provide written informed consent. If writing, did it receive a waiver of consent from the REC?P: 11- In qualitative phase informed consent obtained was verbal due to in this way participants feel more intimate relationship for participating in interview sessions. The IRB at this site was inclined to this procedure. Thereafter, in the quantitative phase, surveys were completed through Porsline software. At the beginning of inventory written informed consent was attached. 

Line 259:This statement is not clear. if you had talked about the consent process here; this would not be necessary; Electronically, some authors will state that by clicking I agree to participate, is a taken as consent. But have to be given information about the study before accessing the survey: P:11- . At the beginning of the inventory written informed consent was attached and they declared their consent by clicking “I agree to participate”.

---

## [Editor Report · Decision Letter 3]

6 May 2021

PONE-D-20-37513R3

Development and psychometric properties of the Nursing Student Academic Resilience Inventory (NSARI): A Mixed-Method study

PLOS ONE

Dear Dr. ghods,

Thank you for submitting your manuscript to PLOS ONE. After careful consideration, we feel that it has merit but does not fully meet PLOS ONE’s publication criteria as it currently stands. Therefore, we invite you to submit a revised version of the manuscript that addresses the points raised during the review process.

In particular, see below some points raised by this Academic Editor.

1) Line 57 “Previous studies conducted in Iran showed that nursing students are affected by various stressors.” You should cite at least a few examples of such studies.

2) Line 70 “In this study, Stephen's resilience model as protective model was 70 used.” You should cite a reference here where the reader can read about Stephen’s model.

3) Lines 202-204: the issue of the modification indices is a result already reported in the Results paragraph lines 319-320.

4) Please comment the under-threshold values of CFI and AGFI obtained in confirmatory factor analysis.

5) When citing a reference, you should use the square parentheses method: line 79 Chmitorz et al. [5] not Chmitorz et al. (2018). See also citations at lines 365, 371, 386, 389, 397, and 418.

6) English editing is still needed: line 126 “regarding to”; lines 139-140; line 155; line 248; line 288 “participates”; line 370 “These results supported the result of the present study.”

We look forward to receiving your revised manuscript.

Kind regards,

Paola Gremigni, Ph.D.

Academic Editor

PLOS ONE

Journal Requirements:

Additional Editor Comments (if provided):

Dear Authors,

there are still points to be improved.

1) Line 57 “Previous studies conducted in Iran showed that nursing students are affected by various stressors.” You should cite at least a few examples of such studies.

2) Line 70 “In this study, Stephen's resilience model as protective model was 70 used.” You should cite a reference here where the reader can read about Stephen’s model.

3) Lines 202-204: the issue of the modification indices is a result already reported in the Results paragraph lines 319-320.

4) Please comment the under-threshold values of CFI and AGFI obtained in confirmatory factor analysis.

5) When citing a reference, you should use the square parentheses method: line 79 Chmitorz et al. [5] not "Chmitorz et al. (2018)". See also citations at lines 365, 371, 386, 389, 397, and 418.

6) English editing is still needed: line 126 “regarding to”; lines 139-140; line 155; line 248; line 288 “participates”; line 370 “These results supported the result of the present study.”

---

## [Author Response · Author response to Decision Letter 3]

12 May 2021

Line 57 “Previous studies conducted in Iran showed that nursing students are affected by various stressors.” You should cite at least a few examples of such studies. 

1. Pourafzal F, Seyedfatemi N, Inanloo M, Haghani H. Relationship between Perceived Stress with Resilience among Undergraduate Nursing Students. Hayat. 2013;19(1):1-12(Persian).

2. Moaddeli Z, Ghazanfari Hesamabadi M. A survey on the students’ exam anxiety in the Fatemeh (PBUH) College of Nursing and Midwifery, Spring 2004. Strides in Development of Medical Education. 2005;1(2):65-72.

3. Nikanjam R, Barati M, Bashirian S, Babamiri M, Fattahi A, Soltanian A. Student-Life Stress Level and its Related Factors among Medical Students of Hamadan University of Medical Sciences in 2015. Journal of Education and Community Health. 2016;2(4):57-64. doi: https://doi.org/10.21859/jech-02048.

2) Line 70 “In this study, Stephen's resilience model as protective model was 70 used.” You should cite a reference here where the reader can read about Stephen’s model.P:3 

In this study, Stephen's resilience model[4]

3) Lines 202-204: the issue of the modification indices is a result already reported in the Results paragraph lines 319-320.It is deleted from method section: 

Regarding the modification indices, 4 pairs of measurement errors among the measured items of the first, third, fourth, and fifth factors were allowed to co-vary freely, which consequently improved the final model fit noticeably

4) Please comment the under-threshold values of CFI and AGFI obtained in confirmatory factor analysis.P:23 

Moreover, a number of fit indices were found to be under the recommended cut-off point. To the best of our knowledge, the NSARI is the first inventory to measure resilience in a theoretically driven way that has the potential to track and predict resilience in nursing students, because of evidences of validity and reliability in this population. To ensure that the instrument is psychometrically robust, its psychometric properties should be tested using different statistical methods such as the Rasch model and structural equation modeling, in order to estimate age and gender effects. Further studies are recommended to investigate the conceptual structure of this inventory to gather more evidences regarding the psychometric properties of the inventory study

5) When citing a reference, you should use the square parentheses method: line 79 Chmitorz et al. [5] not Chmitorz et al. (2018). See also citations at lines 365, 371, 386, 389, 397, and 418.They are revised and highlighted.

6) English editing is still needed: line 126 “regarding to”; lines 139-140; line 155; line 248; line 288 “participates”; line 370 “These results supported the result of the present study.”The article is edited again by editors at NativeEnglishEdit.com and highlighted.

---

## [Editor Report · Decision Letter 4]

17 May 2021

Development and psychometric properties of the Nursing Student Academic Resilience Inventory (NSARI): A Mixed-Method study

PONE-D-20-37513R4

Dear Dr. ghods,

We’re pleased to inform you that your manuscript has been judged scientifically suitable for publication and will be formally accepted for publication once it meets all outstanding technical requirements.

Kind regards,

Paola Gremigni, Ph.D.

Academic Editor

PLOS ONE
---

## [Editor Report · Acceptance letter]

19 May 2021

PONE-D-20-37513R4 

Development and Psychometric Properties of the Nursing Student Academic Resilience Inventory (NSARI): A Mixed-Method Study 

Dear Dr. Ghods:

I'm pleased to inform you that your manuscript has been deemed suitable for publication in PLOS ONE. Congratulations! Your manuscript is now with our production department. 

Kind regards, 

on behalf of

Prof. Paola Gremigni 

Academic Editor

PLOS ONE